# Vortex clustering, polarisation and circulation intermittency in classical and quantum turbulence

Juan Ignacio Polanco [1,2✉], Nicolás P. Müller [1✉] & Giorgio Krstulovic [1✉]

The understanding of turbulent flows is one of the biggest current challenges in physics, as no first-principles theory exists to explain their observed spatio-temporal intermittency. Turbulent flows may be regarded as an intricate collection of mutually-interacting vortices. This picture becomes accurate in quantum turbulence, which is built on tangles of discrete vortex filaments. Here, we study the statistics of velocity circulation in quantum and classical turbulence. We show that, in quantum flows, Kolmogorov turbulence emerges from the correlation of vortex orientations, while deviations—associated with intermittency—originate from their non-trivial spatial arrangement. We then link the spatial distribution of vortices in quantum turbulence to the coarse-grained energy dissipation in classical turbulence, enabling the application of existent models of classical turbulence intermittency to the quantum case. Our results provide a connection between the intermittency of quantum and classical turbulence and initiate a promising path to a better understanding of the latter.

[1] Université Côte d'Azur, Observatoire de la Côte d'Azur, CNRS, Laboratoire J. L. Lagrange, Boulevard de l'Observatoire CS 34229 - F 06304 NICE Cedex 4, Paris, France. [2] Univ Lyon, CNRS, École Centrale de Lyon, INSA Lyon, Univ Claude Bernard Lyon 1, LMFA, UMR5509, 69130 Écully, France.
✉email: juan-ignacio.polanco@ec-lyon.fr; nmuller@oca.eu; krstulovic@oca.eu

Vortices are manifestly the most attractive feature of fluid flows occurring in the Nature. They are highly rotating zones of the fluid that often take the form of elongated filaments, of which tornadoes are one prominent example in atmospheric flows. Such structures can travel and interact with other vortex filaments, as well as with the surrounding fluid. In fact, the dynamics of vortex filaments in fluid flows is highly non-trivial, as they can reconnect changing the topology of the flow[1]. Their non-trivial arrangements may lead to very complex configurations and in particular to turbulence, an out-of-equilibrium state characterised by a large-scale separation between the scales at which energy is injected and the one at which it is dissipated. In three-dimensional flows, because of the inherently non-linear character of turbulence, energy initially injected at large scales is transferred towards the small scales through a cascade-like process.

In turbulent flows, the typical thickness of a vortex filament is comparable to the smallest active scale of turbulence[2], itself usually much smaller than the eddies carrying most of the energy content of the flow. Vortex filaments may thus be seen as the fundamental structure of turbulence, whose collective dynamics leads to the multi-scale complexity of such flows. Indeed, depending on their individual intensities and orientations, a set of vortex filaments located within a given spatial region may contribute constructively or destructively to the fluid rotation rate. In fluid dynamics, the rotation rate of a two-dimensional fluid patch is commonly quantified by the velocity circulation around the closed loop $\mathcal{C}$ surrounding the patch,

$$\Gamma(\mathcal{C}; \mathbf{v}) = \oint_{\mathcal{C}} \mathbf{v} \cdot d\mathbf{r}, \tag{1}$$

where $\mathbf{v}$ is the fluid velocity field. Note that, by virtue of Stokes' theorem, the circulation is equal to the flux of vorticity, $\boldsymbol{\omega} = \nabla \times \mathbf{v}$, through the fluid patch.

The above view of vortex filaments as the fundamental unit of fluid flows is particularly appropriate in superfluids, such as low-temperature liquid helium and Bose–Einstein condensates (BECs). Indeed, in such fluids, vortices are well-defined discrete objects about which the circulation is quantised, taking values multiple of $\kappa = h/m$. Here $h$ is Planck's constant and $m$ is the mass of the bosons constituting the superfluid[3]. Such property arises from their quantum nature, as vortices are topological defects of the macroscopic wave function describing the system. For this reason, vortex filaments in superfluids are called quantum vortices.

One of the most striking properties of low-temperature superfluids is their total absence of viscosity. Despite this fact, quantum vortex reconnections are possible, since Helmholtz' theorem that forbids reconnections in classical inviscid fluids[1] breaks down due to the vanishing fluid density at the vortex core. This picture was first suggested by Feynman in 1955[4] and later confirmed numerically in the framework of the Gross–Pitaevskii (GP) equation[5]. Since then, quantum vortex reconnections have been observed experimentally in superfluid helium[6] and in BECs[7]. They are characterised by universal scaling laws[8,9] and have been linked to irreversibility, both in experiments[10] and in numerical simulations[11]. In the early vortex filament simulations by Schwarz[12], it was noticed that quantum vortex reconnections are a key physical process for the development of quantum turbulence, a state described by the complex interaction of a tangle of quantum vortices. Such a state is illustrated by the vortex filaments (in green and yellow) visualised in Fig. 1, obtained from the GP simulations performed in Ref. 13.

Quantum turbulence is characterised by a rich multi-scale physics. At small scales, between the vortex core size (about 1 Å in superfluid ⁴He) and the mean inter-vortex distance $\ell$ (~1 μm),

the physics is governed by the dynamics of individual quantised vortices[14]. At such scales, Kelvin waves (waves propagating along vortices) and vortex reconnections are the main physical processes carrying energy along scales[15,16]. In contrast, at scales larger than $\ell$, the quantum nature of the superfluid becomes less important and a regime comparable to classical turbulence emerges. Indeed, at such scales, a Kolmogorov turbulent cascade is observed, provided that a large-scale separation exists between $\ell$ and the largest scale of the system. In particular, the scaling law predicted by Kolmogorov's celebrated K41 theory[17] for the kinetic energy spectrum has been observed in superfluid helium experiments[18,19] and in numerical simulations of quantum turbulence[20–22].

Previous studies have suggested that, in quantum turbulence, the emergence of K41 scaling laws is associated to a local *polarisation* of the vortex tangle[14,23–27]. In other words, within a given spatial region, the orientations of nearby vortices are not independent, but instead have some degree of correlation. This phenomenon is visible in Fig. 1, where vortex bundles—regions of same-coloured vortex filaments—can be clearly identified. This local polarisation is present even in ideally isotropic flows, and should not be confused with the preferential large-scale orientation of vortices, which typically occurs in anisotropic flows. A classic example of the latter is a rotating cylindrical vessel filled with superfluid helium[4].

In a recent work[13], we have shown that the quantitative similarities between classical and quantum turbulence go far beyond the Kolmogorov energy spectrum. Indeed, both systems display the emergence of extreme events that result in the breakdown of Kolmogorov's K41 theory—a phenomenon known as intermittency. Our work was motivated by the recent study of Iyer et al.[28], which suggested that intermittency has a relatively simple signature on the statistics of circulation in classical turbulence. In particular, the moments of the circulation measured over fluid patches of area $A \sim r^2$ follow a power law of the form

$$\langle |\Gamma|^p \rangle_A \sim r^{\lambda_p} \sim A^{\lambda_p/2}, \tag{2}$$

with scaling exponents $\lambda_p$ that increasingly deviate from the K41 prediction $\lambda_p^{K41} = 4p/3$ as the moment order $p$ increases. By performing simulations of a generalised GP equation, we have shown that the anomalous scaling exponents $\lambda_p$ in the inertial scales of quantum turbulence closely match those observed in classical turbulence[13]. Note that, up until now, most of the advances in the understanding of intermittency have been made in terms of velocity increments. However, despite many theoretical efforts[17,29–31], there is still no first-principles theory able to explain this phenomenon. The above-cited findings suggest that circulation may provide an alternative path towards a better understanding of turbulence (as first hinted the by pioneering theoretical work of Migdal[32]), and eventually, to novel circulation-based theories of intermittency[33,34].

The strong similarity between the statistics of circulation in classical and quantum turbulence is particularly striking given the very different nature of vortices in both types of fluids. This statistical equivalence opens the way for an interpretation of the intermittency of classical turbulent flows in terms of the collective dynamics of discrete vortex filaments carrying a fixed circulation. With this idea in mind, we relate in this work the intermittent statistics of velocity circulation in classical and quantum turbulence. We start by investigating in quantum turbulence how local vortex polarisation, as well as the non-trivial spatial distribution of vortex filaments, affect circulation statistics. We address the following questions: Is it possible to study both effects separately? Do they contribute in the same way to the flow intermittency? We then provide a relation between the spatial distribution of discrete

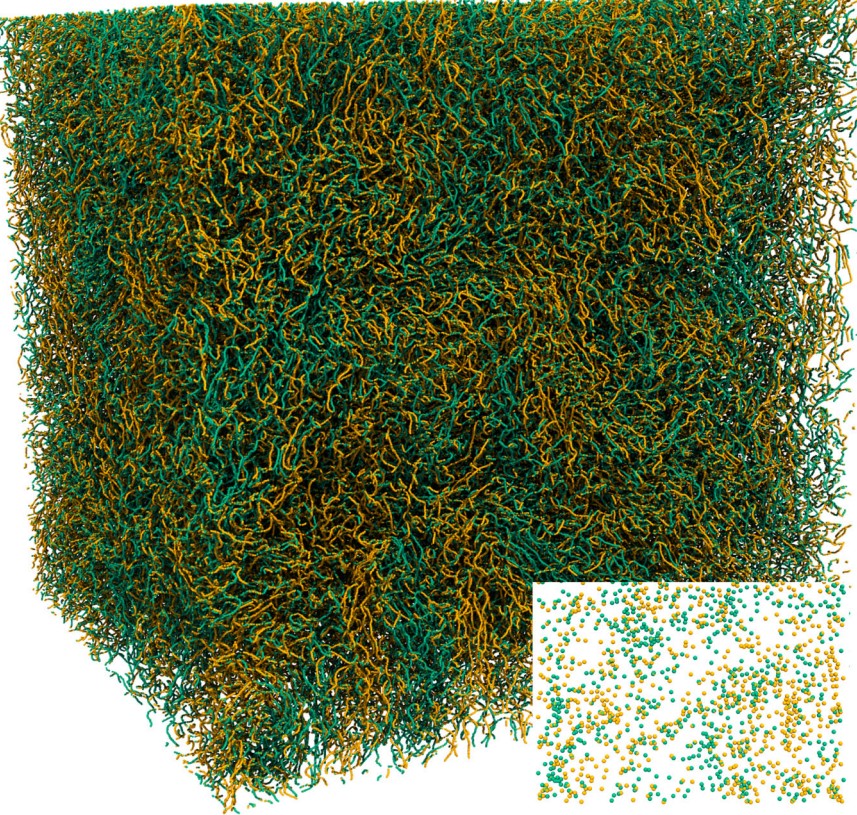

**Fig. 1 Visualisation of a quantum turbulent vortex tangle.** Instantaneous state obtained from GP simulations using $2048^3$ collocation points. Green and yellow colours correspond to opposite orientations of the vortex lines with respect to the vertical direction. The inset shows a horizontal two-dimensional cut of the system. See "Methods" for the vortex identification algorithm.

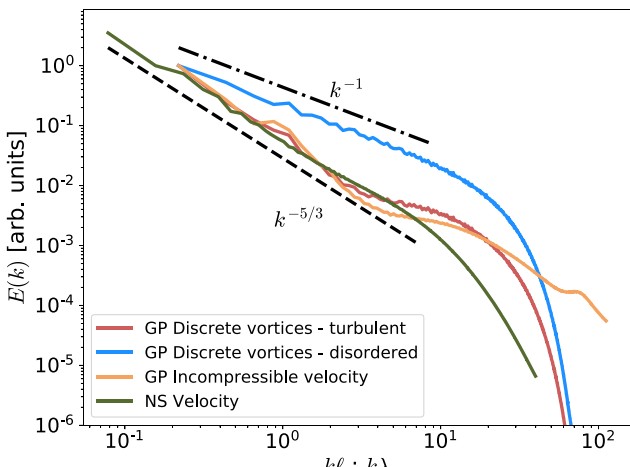

**Fig. 2 Kinetic energy spectrum in quantum and classical turbulence.** Spectra are obtained from simulations of the generalised Gross–Pitaevskii (GP) model and the incompressible Navier–Stokes (NS) equations. Wave numbers $k$ are, respectively, normalised by the mean inter-vortex distance $\ell$ and by the Taylor micro-scale $\lambda$, while the vertical axis is in arbitrary units. In the GP case, the incompressible part of the kinetic energy is plotted[63]. Also shown are the energy spectra obtained after applying the vortex detection procedure to the GP fields (see "Methods" for details), both before and after the randomisation of the vortex orientations (turbulent and disordered cases, respectively). Dashed and dash-dotted lines, respectively, represent the Kolmogorov scaling $k^{-5/3}$ and the disordered scaling $k^{-1}$. Source data are provided as a Source Data file.

vortices, and the coarse-grained energy dissipation rate in classical turbulence, a quantity at the core of existent intermittency models.

In this work, quantum and classical turbulent systems are, respectively, studied using high-resolution direct numerical simulations of a generalised GP and the incompressible Navier–Stokes (NS) equations. Discrete vortices and their signs are extracted from the GP fields and then analysed. To disentangle the effects of polarisation and spatial vortex distribution, we additionally study a disordered turbulence state. Such state is generated from the discrete vortex data by randomly resetting the sign of each individual vortex while keeping its position fixed. To illustrate the differences between the turbulent (non-disordered) and the disordered turbulence states, we plot in Fig. 2 the kinetic energy spectrum associated to each vortex configuration (see "Methods" for details on the computation of the spectra from discrete vortices). First, we see that the turbulent case displays a clear $k^{-5/3}$ range, in agreement with the energy spectra obtained from the full GP and NS fields. Note that, in the case of GP fields, we show the incompressible kinetic energy spectrum, which contains 86% of the total energy of the system— the other components being the compressible, internal and quantum energy[20,22]. Secondly, the K41 scaling disappears once polarisation is artificially suppressed from the tangle, leading to a trivial $k^{-1}$ scaling range for the disordered state (see "Methods" for a brief derivation). Note that this same scaling has already been observed in vortex filament simulations, once the vortex tangle has been decomposed into polarised and random components[26].

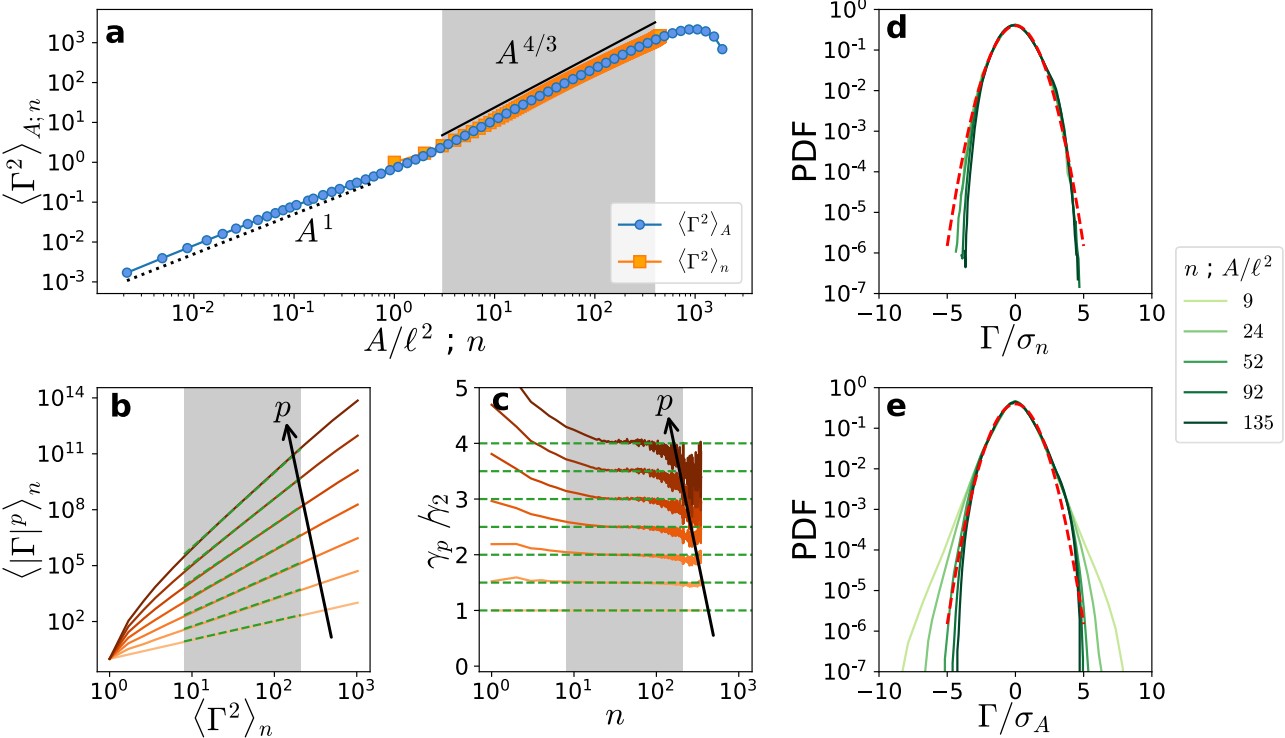

**Fig. 3 Circulation statistics in quantum turbulence. a** Circulation variance as a function of the area $A$ of each loop and of the number $n$ of neighbouring vortices. The dotted line represents the scaling $A^1$ observed at small scales of quantum turbulence[13]. **b, c** Moments and local slopes of $\langle |\Gamma|^p \rangle_n$ as a function of $\langle \Gamma^2 \rangle_n$ according to the ESS approach, for $p \in [2, 8]$. Dashed green lines represent the K41 scaling $\gamma_p = 2p/3$. **d, e** PDFs of the circulation for different (**d**) numbers of vortices and (**e**) loop areas. All PDFs are normalised by the respective standard deviations. Dashed red lines represent a unit Gaussian distribution.

## Results

**A simple discrete model of circulation**. Let us first consider a set of $n$ discrete vortices, each of them carrying a circulation $\kappa s_i$, where $s_i = \pm 1$ is the sign of each vortex. From now on we set the quantum of circulation to $\kappa = 1$ for simplicity. We propose to model the total circulation of the $n$-vortex collection, $\Gamma_n = \sum_{i=1}^{n} s_i$, as a biased one-dimensional random walk. Polarisation is naturally introduced by letting each random step $s_i$ be positively correlated with the instantaneous position $\Gamma_{i-1}$, i.e. the total circulation of all previous vortices.

Concretely, we construct inductively the following toy model for the circulation. The sign of the first vortex, $s_1$, has equal probability of being positive or negative. Then, the sign of vortex $n + 1$ is positive with a probability $p_{n+1}$, which we set to depend on the total circulation at step $n$ as $p_{n+1} = [1 + f(\Gamma_n/n)]/2$. Here, $f(z)$ is a suitable function (odd, non-decreasing, taking values in $[-1, 1]$), such that $p_n \in [0, 1]$ at each step $n$. For the sake of simplicity, we choose here $f(z) = \beta z$ (see the Supplementary information for the general case), where $\beta \in [0, 1]$ is an adjustable parameter that sets the polarisation of the system. When $\beta = 0$, one retrieves a standard random walk with scaling $\langle |\Gamma|^2 \rangle_n \sim n$. Conversely, for $\beta = 1$, one recovers a fully polarised set of vortices behaving as $\langle |\Gamma|^2 \rangle_n \sim n^2$.

The resulting model is a discrete Markov process, since the probability distribution of $\Gamma_{n+1}$ only depends on the state $\{n, \Gamma_n\}$ via the probability $p_{n+1}$. Concretely, the probability $\mathcal{P}_n(\Gamma)$ of having $\Gamma_n = \Gamma$ obeys the master equation

$$\mathcal{P}_{n+1}(\Gamma) = \left( \frac{1}{2} + \beta \frac{\Gamma - 1}{2n} \right) \mathcal{P}_n(\Gamma - 1) + \left( \frac{1}{2} - \beta \frac{\Gamma + 1}{2n} \right) \mathcal{P}_n(\Gamma + 1). \tag{3}$$

Multiplying this equation by $\Gamma^2$, summing over all $\Gamma$ and, for the sake of simplicity, taking the limit of continuous $n$, one gets a closed equation for the circulation variance,

$$\frac{d \langle \Gamma^2 \rangle_n}{dn} = 1 + \frac{2\beta}{n} \langle \Gamma^2 \rangle_n, \tag{4}$$

where averages are over all realisations after $n$ steps. For large $n$, this equation predicts the scaling $\langle \Gamma^2 \rangle_n \sim n$ for $\beta < 1/2$ (corresponding to a set of vortices with negligible polarisation), and $\langle \Gamma^2 \rangle_n \sim n^{2\beta}$ otherwise. In particular, choosing $\beta = 2/3$, one recovers the Kolmogorov scaling by replacing $n \propto A$. This relation between the number of vortices $n$ and the loop area $A$ containing them is expected to hold on average under spatial homogeneity conditions, but neglects potentially important inhomogeneities in the spatial vortex distribution that may affect high-order moments of $n$. Besides, for the $p$th order moment, the model predicts the self-similar scaling $\langle |\Gamma|^p \rangle_n \sim n^{\gamma_p}$ with $\gamma_p = \beta p$. More generally, for any suitable function $f(z)$ defining the probability $p_n$ of the model, one obtains the linear scaling $\gamma_p = p \min\{\max\{1/2, f'(0)\}, 1\}$ (see the Supplementary information for more details on the calculations).

The toy model introduced above shows in a very simple manner how a specific correlation (or polarisation) is responsible for the emergence of non-trivial scaling laws, as already suggested by previous works on quantum turbulence[24–26]. In addition, the model yields self-similar statistics, suggesting that polarisation is not sufficient to reproduce the observed intermittency of circulation in classical[28] and quantum[13] turbulent flows. At this point, we may speculate that the lack of intermittency in the model is likely associated with the missing notion of space. Indeed, on average one expects to have a number of vortices $\langle n \rangle \sim A/\ell^2$ crossing a loop of area $A$, where $\ell$ is the mean inter-

vortex distance. Yet, fluctuations in their spatial distribution—associated to the appearance of vortex clusters and voids—may strongly influence high-order moments. As seen in Fig. 1, such effects clearly take place in turbulent flows, where they are linked to the formation of coherent structures.

**Comparison with quantum turbulence data**. The ideas hinted at by our toy model can be verified using actual quantum turbulence data. With this aim, we identify all vortex filaments present in our GP simulations (see "Methods" for details), and compute circulation statistics as a function of the number $n$ of considered vortices. Crucially, groups of vortices are chosen based on their spatial proximity, which is required to preserve the correlation between vortices. On the other hand, with such a conditioning, one may expect the effect of strong spatial fluctuations of the vortex distribution to be somewhat relaxed. In practice, for each two-dimensional cut of the simulation, we consider sets of $n$ neighbouring vortices in order to compute the circulation moments $\langle |\Gamma|^p \rangle_n$. Then, to improve the statistics, we repeat such measurement for each cut and along the three Cartesian directions.

The resulting second-order moment $\langle \Gamma^2 \rangle_n$ is shown in Fig. 3a, along with the moment $\langle \Gamma^2 \rangle_A$ measured for different loop areas $A$ (data from Müller et al.[13]). At small scales, $\langle \Gamma^2 \rangle_A \sim A^1$ due to the discrete nature of vortices[13]. In contrast, within the inertial range, both moments clearly exhibit the expected Kolmogorov scaling. In particular, $\langle \Gamma^2 \rangle_n \sim n^{\gamma_2}$ with $\gamma_2 = 4/3$. This result allows us to use the extended self-similarity (ESS) framework[35] to determine the scaling properties of higher-order moments via the relation $\langle |\Gamma|^p \rangle_n \sim n^{\gamma_p} \sim \langle \Gamma^2 \rangle_n^{\gamma_p/\gamma_2}$. Remarkably, as shown in Fig. 3b–c, the moments display a clear self-similar behaviour with $\gamma_p = 2p/3$, thus obeying Kolmogorov scaling for all orders. The self-similarity is also observed in the normalised probability density functions (PDFs) of $\Gamma$ for different values of $n$ (Fig. 3d), which nearly collapse and are close to Gaussian. This behaviour should be contrasted with the non-collapsing PDFs of $\Gamma$ for different loop areas $A$ (Fig. 3e). Note that, in both cases, the chosen values of $A$ and $n$ lay within the inertial range, represented by a grey background in Fig. 3a–c.

**Disentangling polarisation and spatial vortex distribution**. The fitted scaling exponents for the turbulent case $2\gamma_p^{\text{turb}}$, discussed above, are plotted in Fig. 4 (blue-filled stars) as a function of the moment order $p$. These exponents are compared to the measured values of $\lambda_p^{\text{turb}}$ (blue-filled circles) obtained according to Eq. (2), where averages are performed for different loop areas $A$. The latter are the same as in Ref. [13]. The factor 2 in front of $\gamma_p^{\text{turb}}$ comes from considering the relation $\langle n \rangle \sim A \sim r^2$. As discussed earlier, the moments averaged for different $n$ closely follow the self-similar K41 scaling $2\gamma_p^{\text{turb}} \approx 4p/3$ (blue solid line), while the $\lambda_p^{\text{turb}}$ exponents—affected by the spatial vortex distribution—show signs of intermittency[13].

To further distinguish the effects of polarisation and spatial vortex distribution on circulation statistics, we perform the following numerical experiment. We recompute the circulation in the quantum turbulent flow, but before doing this, we randomise the sign of each vortex on each analysed two-dimensional cut while keeping its position fixed. By doing this, we get rid of the system polarisation, while maintaining the non-trivial spatial distribution of vortices. We refer to this system as disordered turbulence. In our non-intermittent toy model, this setting would correspond to the unpolarised value $\beta = 0$, yielding the self-similar circulation scaling $\langle |\Gamma|^p \rangle_n \sim n^{p/2}$. In Fig. 4, we display the

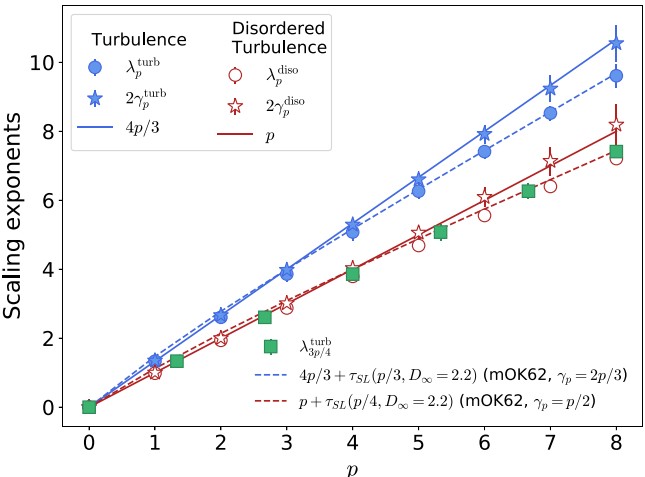

**Fig. 4 Velocity circulation scaling exponents.** Exponents of the turbulent (in blue) and the disordered turbulence (in red) cases. For each case, the scaling exponents are defined as $\langle |\Gamma|^p \rangle_A \sim r^{\lambda_p}$ (circles) and $\langle |\Gamma|^p \rangle_n \sim n^{\gamma_p}$ (stars). Error bars indicate 95% confidence intervals. Self-similar predictions for each case are shown as solid lines. Green squares show the relation between the turbulent and disordered systems given by Eq. (5). The blue and red dashed lines show the OK62 prediction combined with the She–Lévêque model (8) with $D_\infty = 2.2$ (termed "mOK62") for turbulence and disordered turbulence, respectively. Source data are provided as a Source Data file.

corresponding measured exponents of the disordered state $\lambda_p^{\text{diso}}$ and $2\gamma_p^{\text{diso}}$ (red unfilled markers). Remarkably, even after suppressing vortex polarisation, $\lambda_p^{\text{diso}}$ also presents intermittency deviations. In contrast, the scaling exponents $\gamma_p^{\text{diso}}$ satisfy the expected self-similar behaviour $2\gamma_p^{\text{diso}} \approx p$ (red solid line).

The previous results suggest that the non-trivial polarisation of vortices, while being responsible for Kolmogorov scalings, has no major influence on the intermittency of the system. Furthermore, they indicate that the latter originates from fluctuations of the spatial distributions of vortices. From our above observations, one may therefore expect the scaling exponents of the circulation to be given by a composition of the polarisation and spatial distribution effects. That is, we may conjecture that the scaling exponents $\lambda_p$ and $\gamma_p$ are related by

$$\lambda_p = g(\gamma_p) \qquad (5)$$

where $g$ is some yet unknown function.

In order to check this idea, we can try to relate the scaling exponents of the turbulent and disordered turbulent systems. If relationship (5) were to hold true, one should have that $\lambda_p^{\text{diso}} = \lambda_{3p/4}^{\text{turb}}$. Using this relation with the measured exponents of the turbulent case, one indeed recovers the intermittency exponents $\lambda_p^{\text{diso}}$ of the disordered case, as shown by the green squared markers in Fig. 4. This result strongly highlights the importance of the fluctuations of vortex concentration on the intermittency of circulation.

**Spatial vortex distribution and OK62 theory**. As a first step towards relating the intermittency of classical and quantum turbulence, we now quantify the spatial distribution of vortices in the latter system. If vortices were homogeneously distributed in space, then the number $n$ of vortices within loops of area $A$ would be expected to follow a Poisson distribution with mean value $\langle n \rangle_A \propto A$. In that case, the moments of $n$ would scale as $\langle n^p \rangle_A \sim$

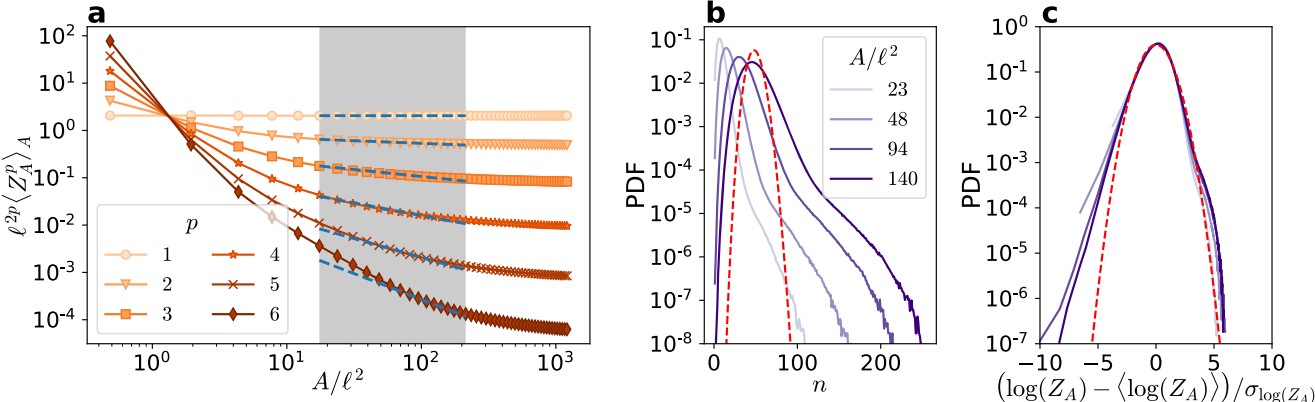

**Fig. 5 Statistics of spatial vortex distribution in quantum turbulence. a** Moments of the number of vortices normalised by the area that contains them, $Z_A = n/A$. Dashed lines correspond to the scaling of Eq. (7), using the She–Lévêque prediction[44] for the anomalous exponents $\tau(p)$ with $D_\infty = 1$. **b** PDFs of the number of vortices contained in loops of varying area $A/\ell^2$. The dashed line corresponds to a Poisson distribution of mean equal to $\langle n \rangle_A$ at $A = 140\ell^2$. **c** Centred reduced PDFs of $\log(Z_A)$ for different values of $A/\ell^2$. A log-normal distribution is shown as reference (red dashed line).

$A^p$ for sufficiently large $A$. Equivalently, the number of vortices per unit area $Z_A = n/A$ would follow the trivial scalings $\langle Z_A^p \rangle_A \sim 1$ for all $p > 0$. As shown in Fig. 5a, this is clearly not the case, indicating that the spatial distribution of vortices is non-trivial in quantum turbulence (as may be inferred from the visualisation of Fig. 1). Indeed, while the first-order moment recovers a constant (consistently with the relation $\langle n \rangle_A \sim A$), higher-order moments of $Z_A$ follow a different scaling with a negative exponent—a sign of anomalous behaviour. This is confirmed by the PDFs of $n$ displayed in Fig. 5b, which are long-tailed and strongly differ from a Poisson distribution (dashed line).

In classical turbulence, it is today well accepted that the intermittency of velocity fluctuations is linked to the emergence of violent events, characterised by strong spatial fluctuations of the kinetic energy dissipation rate $\varepsilon(\mathbf{x})$. Such idea led Obukhov and Kolmogorov in 1962 to develop a refined similarity theory of turbulence, commonly referred to as OK62 theory, where such fluctuations are taken into account[17,36,37], unlike K41 theory which only deals with the mean value of $\varepsilon(\mathbf{x})$. This refined theory considers the scale-averaged (or coarse-grained) energy dissipation rate $\varepsilon_r(\mathbf{x}) = \frac{3}{4\pi r^3} \int_{B(\mathbf{x},r)} \varepsilon(\mathbf{x}') \, d^3\mathbf{x}'$, where $B(\mathbf{x}, r)$ is a ball of radius $r$ centred at $\mathbf{x}$. When applied to the spatial velocity increments $\delta v_r$ over a distance $r$, OK62 theory states that the statistics of $\delta v_r/(\varepsilon_r r)^{1/3}$ is self-similar and universal. Most intermittency models use $\varepsilon_r$ to predict the anomalous scaling of velocity increment statistics[17]. Some early experiments in classical turbulence showed that, when velocity increments are conditioned on the coarse-grained dissipation, their statistics becomes Gaussian[38,39], proving that the intermittency of velocity fluctuations is hidden behind the distribution of energy dissipation. This observation was later confirmed by numerical simulations[40].

In the case of low-temperature quantum turbulence, such as the one studied here, energy is taken away from the inertial range and transferred towards small scales by the Kelvin wave cascade and vortex reconnections[9,41]. Furthermore, the velocity field diverges at the vortex core, and thus the definition of the dissipation field is delicate. Nevertheless, we can give a phenomenological interpretation of the dissipation by assuming that the system is well represented as a dilute point-vortex gas. Such a picture was recently used by Apolinário et al.[33] to model the velocity circulation in classical turbulence, and becomes particularly pertinent in quantum fluids. Although the superfluid is inviscid, one can model small-scale physics by some effective viscosity $\nu_{\mathrm{eff}}$[23,42], whose value is not important here. This approach allows us to directly estimate the coarse-grained

dissipation field by using its classical definition in terms of velocity gradients and a Dirac-like supported vorticity field (see Supplementary information). Given a disk of radius $r$ crossed by $n$ vortices, a straightforward calculation gives the estimate

$$\varepsilon_r \sim \frac{\nu_{\mathrm{eff}} \kappa^2}{\xi^2} \frac{n}{A} = \frac{\nu_{\mathrm{eff}} \kappa^2}{\xi^2} Z_A, \qquad (6)$$

where $\varepsilon_r$ is the average of the local dissipation rate $\varepsilon(\mathbf{x})$ over the disk, $A = \pi r^2$ is the disk area, $\xi$ the typical vortex thickness and $\kappa$ the quantum of circulation. The number of vortices per unit area $Z_A = n/A$ would then be the quantum analogous of the coarse-grained dissipation $\varepsilon_r$. Remarkably, and similarly to $\varepsilon_r$—which is known to exhibit log-normal statistics in classical turbulence[43]—the normalised PDFs of $\log(Z_A)$ almost collapse and are close to Gaussian in the bulk (Fig. 5c), reinforcing the pertinence of relation (6).

To make a stronger connection between classical and quantum turbulence, we recall that the classical coarse-grained energy dissipation rate is a highly fluctuating quantity that presents anomalous scaling laws traditionally denoted by $\langle \varepsilon_r^p \rangle \sim r^{\tau(p)}$. It follows from Eq. (6) that the number of vortices should satisfy

$$\langle n^p \rangle_A \sim r^{\alpha(p)} \sim A^{\alpha(p)/2}, \quad \text{with} \ \alpha(p) = 2p + \tau(p). \qquad (7)$$

Note that, because of homogeneity, $\tau(1) = 0$, which translates as $\langle n \rangle_A \sim A$ for the mean number of vortices. In the classical turbulence literature, there are several multifractal models for the anomalous exponents $\tau(p)$ that are able to reproduce experimental and numerical measurements[17]. Among those, the She–Lévêque model[44]

$$\tau_{\mathrm{SL}}(p) = -2p/3 + (3 - D_\infty)\left[1 - \left(\frac{7/3 - D_\infty}{3 - D_\infty}\right)^p\right] \qquad (8)$$

has one adjustable parameter $D_\infty$ corresponding to the fractal dimension of the most singular structures of the system. In the original model, which closely matches existent turbulence measurements[35,44–46], these structures are assumed to be vortex filaments, hence $D_\infty = 1$. The combination of prediction (7) with the original She–Lévêque model, represented by the blue dashed lines in Fig. 5a, is in good agreement with our quantum turbulence data for sufficiently large $A$, although some deviations due to the limited scaling range may be present.

**Classical turbulence and conditioned circulation.** We now apply some of the previous ideas to classical turbulence. We perform a direct numerical simulation of the NS equations in a

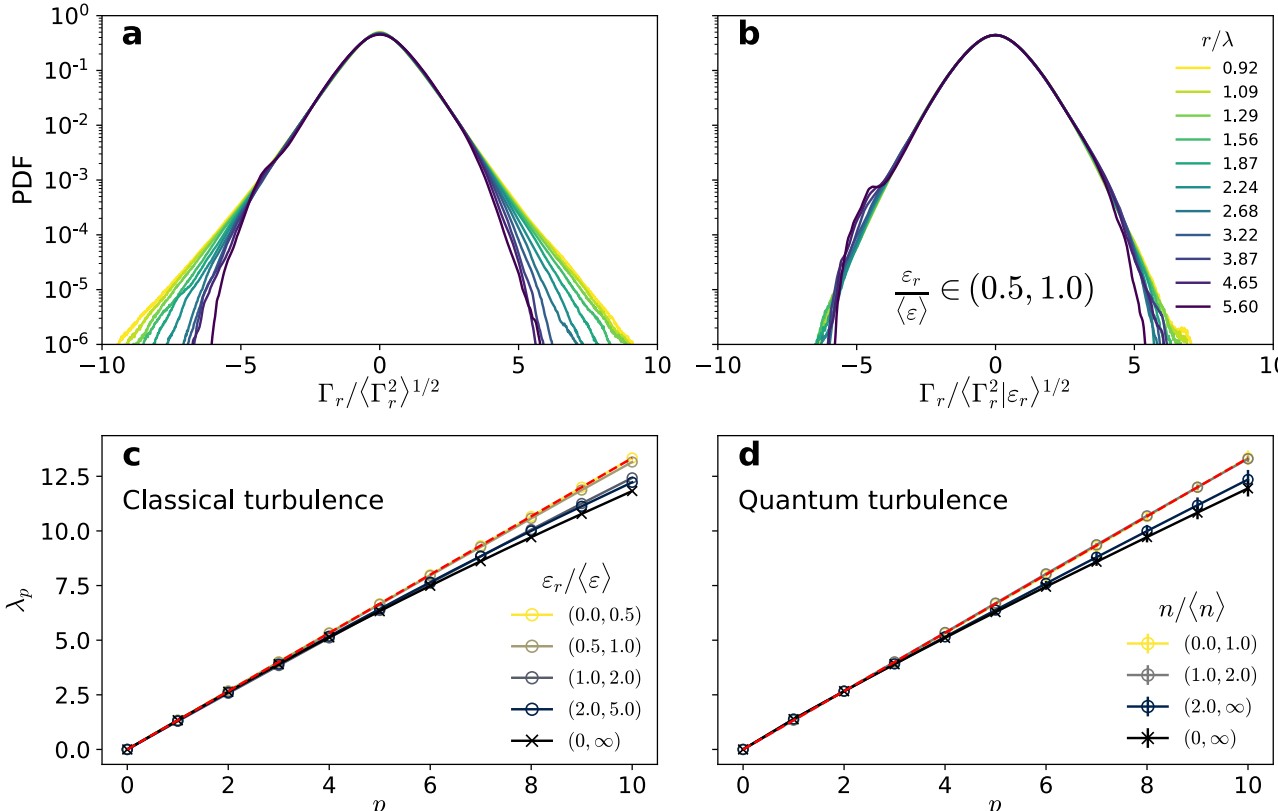

**Fig. 6 Circulation intermittency and OK62 theory in classical and quantum turbulence.** Top panels: PDFs of the circulation in classical turbulence as a function of the loop size $r$. **a** Unconditioned PDFs. **b** PDFs conditioned on low values of the local coarse-grained dissipation, $\varepsilon_r/\langle\varepsilon\rangle \in [0.5, 1]$. The different colours correspond to different loop sizes within the inertial range. **c**, **d** Scaling exponents of the circulation moments in (**c**) classical and (**d**) quantum turbulence. Different colours indicate a conditioning (**c**) on the local dissipation and (**d**) on the number of vortices within each loop. The unconditioned exponents are shown with black crosses. The Kolmogorov self-similar scaling is shown as reference (red dashed line). Error bars indicate 95% confidence intervals.

statistically steady state at a Taylor-based Reynolds number of $\mathrm{Re}_\lambda = 510$. The simulation is performed using $2048^3$ collocation points. We then compute the velocity circulation over planar square loops of area $A = r^2$, and, following the framework of the OK62 refined similarity hypothesis, we condition its statistics on the coarse-grained dissipation field $\varepsilon_r$. The latter is obtained by averaging the local dissipation $\varepsilon$ over the interior of each loop. See "Methods" for details on the numerical simulations and the data analysis.

We first consider the unconditioned velocity circulation PDFs, shown in Fig. 6a. The PDFs display heavy tails (associated with intermittency) which depend on the considered scale $r/\lambda$, with $\lambda$ the Taylor micro-scale. This is consistent with the classical turbulence simulations of Iyer et al.[28,47] The PDF tails are strongly suppressed when the statistics is conditioned on low values of the local coarse-grained dissipation, $\varepsilon_r/\langle\varepsilon\rangle \in [0.5, 1]$, as seen in Fig. 6b. The suppression of intermittency is also manifest in Fig. 6c, where the scaling exponents of circulation are displayed after conditioning on different intervals of $\varepsilon_r$. With no conditioning (black crosses), the scaling exponents match those of Iyer et al.[28], whereas when conditioning on low values of $\varepsilon_r$ the K41 self-similar scaling is recovered.

Note that the above conditioning is slightly different from the one presented in Fig. 3, as here we are conditioning both on the loop area $A$ and on the value of $\varepsilon_r$ within such loops. In the case of quantum turbulence, the equivalent would be to study $\langle\Gamma^p | n\rangle_A$, i.e. to consider only loops of area $A$ having $n$ vortices. Such a double

conditioning is very restrictive, as it requires a very large amount of statistics. Nevertheless, we perform a similar analysis, considering loops having a low, average and high number of vortices relative to the mean. The respective scaling exponents are displayed in Fig. 6d. We find that, for loops with low and average number of vortices, the self-similar K41 scaling is recovered, whereas for loops having large vortex concentrations the statistics is still intermittent. The lack of self-similarity in regions of high dissipation (in classical flows) or high vortex concentration (in quantum flows) hints at the idea that not all such events contribute equally to circulation statistics.

**Can OK62 theory describe circulation intermittency?** Considering the relation introduced in Eq. (6) and the fact that the number of vortices per unit area follows the same intermittent behaviour as $\varepsilon_r$, one could try to apply OK62 theory to relate scaling exponents of circulation $\lambda_p$ with those of dissipation $\tau(p)$, as traditionally done for velocity increments. Within this reasoning, $\Gamma \sim \varepsilon_r^{1/3} r^{4/3}$, yielding a OK62-based relation $\lambda_p = 4p/3 + \tau(p/3)$. However, such a relation is in strong disagreement with our data (classical and quantum turbulence, see Supplementary information) and with early NS studies[48]. Nevertheless, this disagreement is not in contradiction with the fact that the anomalous scaling of the number of vortices is well described by standard multifractal dissipation models (see Fig. 5). Indeed, if one considers a vortex dipole (two vortices of same magnitude and opposite sign), their

contribution to large fluctuations of the local dissipation field and to velocity increments may be very important. On the other hand, for the circulation, the dipole contribution is exactly zero due to vortex cancellation. This fact suggests that not all extreme dissipation events result in extreme circulation values. In particular, intense circulation events would be correlated to those highly dissipative structures in turbulence which carry a strong vortex polarisation, such as vortex sheets or bundles (at scales $r \gg \ell$). Note that, in classical fluids, the idea of vortex filaments organising into groups forming vortex sheets is consistent with the recently proposed sublayers' vortex picture of dissipation[49].

The previous observations motivate us to introduce a modified OK62 theory for the circulation ("mOK62" in the following), where the most relevant singular structures are not vortex filaments but structures of higher fractal dimension. To check this idea, we adapt the She–Lévêque model $\tau_{SL}(p)$ (Eq. (8)) by setting $D_\infty = 2.2$ instead of 1. The chosen dimensionality exactly corresponds to the monofractal fit obtained by Iyer et al.[28] and Müller et al.[13] for the high-order circulation moments ($p > 3$) in classical and quantum turbulence, and, as suggested in the former work, it may be linked to the effect of wrinkled vortex sheets. Note that, for large $p$, our mOK62 model simplifies to $\lambda_p \approx \frac{10}{9}p + (3 - D_\infty)$, which is equivalent to the monofractal fit by Iyer et al.[28]. In Fig. 4, it is shown that the adapted model matches strikingly well the anomalous exponents of circulation both in the turbulent and in the disordered cases for $p > 3$ (dashed lines), while for $p < 3$ there are some deviations.

Our mOK62 model can be generalised to an arbitrary degree of polarisation, which is fully determined by the exponent $\gamma_1 \in [1/2, 1]$. Using dimensional analysis and reintroducing the fundamental quantum of circulation $\kappa$, we have $\Gamma \sim \varepsilon_r^{\gamma_1/2} r^{2\gamma_1} \kappa^{1-3\gamma_1/2}$, leading to $\lambda_p = 2p\gamma_1 + \tau(p\gamma_1/2)$. Accordingly, the conjecture stated in Eq. (5) would be fulfilled with $g(x) = 2x + \tau(x/2)$. We recall that K41 turbulence corresponds to $\gamma_1 = 2/3$, in which case the dependence on $\kappa$ consistently disappears. This model also accurately reproduces disordered turbulence data (see Fig. 4), which corresponds to $\gamma_1 = 1/2$. In this case, $\lambda_p = p + \tau(p/4)$, and intermittency corrections thus vanish at $p = 4$ (instead of $p = 3$ in the turbulent case).

The previous results provide a possible interpretation for the difference between the intermittency of velocity fluctuations and of circulation, based on the different topologies of the dissipative structures contributing to extreme events. We shall notice that an alternative interpretation is also possible, based on the recent works by Apolinário et al.[33] and Moriconi[34]. In this framework, the circulation should scale as $\varepsilon_r^{1/2}$, instead of $\varepsilon_r^{1/3}$, namely $\Gamma \sim \varepsilon_r^{1/2} \nu_r^{-1/2} r^2$, where $\nu_r$ is Kraichnan's eddy viscosity[50]. The latter is found by assuming that the energy spectrum takes the form $E(k) \sim \varepsilon^{2/3} k^{-5/3+\alpha}$ (where $\alpha$ is an intermittency correction), yielding $\nu_r \sim r^{4/3+\alpha}$. Note that this phenomenological approach mixes a mean-field approximation for determining $\nu_r$ with the fluctuations arising from $\varepsilon_r^{1/2}$. Moreover, in its present form, it does not directly account for vortex cancellations. Nevertheless, when combined with the standard She–Lévêque model (with $D_\infty = 1$), this model provides an expression for the exponents $\lambda_p$ as accurate as our mOK62 model in the turbulent case. There is certainly a need to pursue further investigations to understand how both models differ and complement each other.

## Discussion

In this work, we have attempted at providing an interpretation for the intermittent statistics of velocity circulation in turbulent flows. We have done so by viewing turbulent flows as a polarised tangle of discrete and thin vortex filaments, each carrying a constant circulation. While this view is a priori only appropriate in low-temperature quantum fluids, we expect it to be a very pertinent model of classical turbulence, considering the strong similarities recently unveiled between both systems[13].

By introducing and solving a simple toy model and by analysing data of GP quantum turbulence simulations, we have shown that, in discrete-vortex systems, the Kolmogorov self-similar scalings result from a partial polarisation of the vortices (in agreement with previous quantum turbulence studies), while the intermittency of circulation statistics is linked to the non-trivial (non-Poissonian) spatial distribution of vortices. In fact, within fluid patches of varying area $A$ in the inertial range of scales, the number of vortices $n$ is found to be the quantum equivalent of the coarse-grained dissipation $\varepsilon_r$ in classical turbulence, as they both follow the approximately log-normal distribution first hypothesised by the celebrated Obukhov–Kolmogorov OK62 theory for $\varepsilon_r$[43]. Quantitatively, we show that the intermittency of $n$ is well described by the She–Lévêque model for $\varepsilon_r$, confirming the strong equivalence between both observables.

It is important to remark that the quantum turbulence simulations presented in this work have been performed on periodic domains, and are based on the GP equation describing an ideal superfluid at very low temperature. In contrast, most superfluid turbulence experiments using liquid helium are performed in confined systems and at finite temperatures[18,19], in a regime that may be described by a two-fluid model[51]. Early experimental studies showed that the signature of intermittency on velocity increments is nearly independent of the temperature, matching observations in classical fluids[52–54]. These observations were later contradicted by a recent experimental investigation, which showed an enhancement of velocity intermittency in the two-fluid regime compared to classical turbulence[55], in agreement with previous numerical simulations of related models[56,57]. Compared to velocity increments, we expect the circulation to be a much more robust observable in quantum fluids, as it does not display singular behaviour in the vicinity of vortices[13]. For this reason, measuring the scaling properties of circulation in future experiments may help disambiguate existent contradictions, and provide a clearer answer on the intermittency of finite-temperature quantum turbulence. Recent experiments have made initial attempts at reconstructing Eulerian velocity fields from Lagrangian particle tracking measurements in turbulent superfluid helium[55]. Such a technique could be used in principle to measure the velocity circulation in superfluid helium, although addressing high-order statistics might still be challenging. However, note that such an approach is delicate because, due to the two-fluid nature of finite-temperature superfluid helium, particles may fail to capture important Eulerian flow features[58,59], and further work is needed to determine its suitability.

Finally, using data from NS and GP simulations, we have confirmed that the classical OK62 theory does not fully account for the intermittency of the circulation in classical and quantum turbulence. We have provided an explanation based on the presumed topology of the turbulent structures that most contribute to extreme circulation events. We have then proposed a modified OK62 description of circulation, where relevant singular structures have a fractal dimension $D_\infty \approx 2.2$ associated to vortex sheets[28]. This value differs from the dimensionality $D_\infty = 1$ of isolated vortex filaments, used in the modelling of velocity increment statistics[44]. Using this idea, we have shown that the intermittency of circulation is well reproduced by a modified version of the She–Lévêque model, bringing support to the vortex sheet interpretation first proposed by Iyer et al.[28]. All the previous ideas were additionally tested by introducing a disordered turbulence state, obtained by artificially suppressing vortex polarisation from a GP numerical simulation.

There are still some questions that remain open for future works. In particular, further investigation on the topology of relevant structures for the intermittency of circulation is required. We have argued that the smallest structures significant for circulation are vortex sheets, as simpler structures are irrelevant due to vortex cancellation. One way of approaching this topic is by use of cancellation exponents[60–62], method that exploits the fact that circulation can take either negative or positive values. An alternative approach, based on recent works by Apolinário et al.[33] and Moriconi[34], suggests that the most relevant singular structures for velocity circulation should still be vortex filaments. Further investigations on the fractal dimension of circulation would help develop more accurate models of intermittency.

Our findings hint at the existence of a coarse-grained quantity different from $\varepsilon_r$, which may better encapsulate the intermittency of circulation in classical turbulence in the spirit of an OK62-like theory. Furthermore, it may be appropriate to investigate the relevance of quantities, such as the local vorticity magnitude (or enstrophy) or the local strain. Such coarse-grained quantity would be expected to display intermittent statistics with extreme values associated to the presence of quasi-two-dimensional structures such as vortex sheets.

More generally, our present results reinforce the strong equivalence between classical and quantum turbulence, and constitute an attempt at providing an explicit connection between the intermittency of both systems. We expect such a connection to provide a possible path to a simplified description of the intermittency of classical turbulence, a highly challenging topic from a modelling standpoint, yet extremely relevant to the understanding of fluid flows occurring in the Nature.

## Methods

### Numerical simulations

We study the dynamics of quantum turbulence in the framework of a generalised GP model

$$i\hbar \frac{\partial \psi}{\partial t} = -\frac{\hbar}{2m}\nabla^2 \psi - \mu(1+\chi)\psi \\ + g\left(\int V_{\mathrm{I}}(\mathbf{x}-\mathbf{y})\,|\psi(\mathbf{y})|^2\,\mathrm{d}^3 y\right)\psi + g\chi \frac{|\psi|^{2(1+\gamma)}}{n_0^\gamma}\psi, \quad (9)$$

where $\psi$ is the condensate wave function describing the dynamics of a compressible superfluid at zero temperature. Here, $m$ is the mass of the bosons, $\mu$ is the chemical potential, $n_0$ the particle density and $g = 4\pi\hbar^2 a_s/m$ is the coupling constant proportional to the s-wave scattering length. The dimensionless parameters $\chi$ and $\gamma$ correspond to the amplitude and order of beyond mean field corrections. The nonlocal interaction between bosons is given by the potential $V_{\mathrm{I}}(\mathbf{x}-\mathbf{y})$ which is chosen, together with $\chi$ and $\gamma$, to reproduce the roton minimum in the excitation spectrum and the equation of state of superfluid helium. Details on the chosen parameters can be found in Ref. [22]. The use of a standard or a generalised GP model does not affect the statistics of velocity circulation[13].

The hydrodynamic interpretation of Eq. (9) stems from the Madelung transformation $\psi = \sqrt{\rho/m}e^{i m\phi/\hbar}$, where $\rho$ is the local density and $\phi$ the phase of the complex wave function. The velocity field is then given by $\mathbf{v} = \nabla\phi$. Note that $\phi$ is not defined at the locations where $\psi$ vanishes, which implies that the velocity field is singular along quantum vortices[63].

The generalised GP equation (9) is solved in a three-dimensional periodic cube by direct numerical simulations using the Fourier pseudospectral code FROST, with an explicit fourth-order Runge–Kutta method for the time integration[22]. The quantum turbulent regime is studied in a freely decaying Arnold–Beltrami–Childress (ABC) flow[13,21] with $2048^3$ collocation points. To reduce acoustic emissions, the initial condition is prepared using a minimisation process[20]. The box has a size $L = 1365\xi$ and the inter-vortex distance is $\ell \approx 28\xi$, with $\xi$ the healing length.

We also perform direct numerical simulations of the incompressible NS equations

$$\frac{\partial \mathbf{v}}{\partial t} + \mathbf{v}\cdot\nabla\mathbf{v} = -\nabla p + \nu\nabla^2\mathbf{v} + \mathbf{f}, \quad (10)$$

$$\nabla\cdot\mathbf{v} = 0, \quad (11)$$

using the Fourier pseudospectral code LaTu[64] in a periodic cubic domain. The temporal advancement is performed with a third-order Runge-Kutta scheme.

Above, $p$ is the pressure field, $\nu$ the fluid kinematic viscosity and $\mathbf{f}$ an external forcing stirring the fluid. The latter acts at large scales within a spherical shell of radius $|\mathbf{k}| \leq 2$ in Fourier space. The turbulent regime is studied once the simulation reaches a statistically steady state. The simulation is performed using $2048^3$ collocation points at a Taylor-based Reynolds number of $\mathrm{Re}_\lambda = 510$.

### Evaluation of circulation and coarse-grained dissipation

To obtain the circulation from GP and NS simulation data, we take advantage of the spectral nature of both solvers, and compute the circulation from the Fourier coefficients of the velocity fields. Namely, over a given $L$-periodic 2D cut of the physical domain, we write the circulation over a square loop of side $r$, centred at a point $\mathbf{x} = (x, y)$, as the convolution

$$\Gamma_r(\mathbf{x}) = \int_{B_r(\mathbf{x})} \omega(\mathbf{x}')\,\mathrm{d}^2\mathbf{x}' = \iint_{[0,L]^2} H_r(\mathbf{x}-\mathbf{x}')\,\omega(\mathbf{x}')\,\mathrm{d}^2\mathbf{x}', \quad (12)$$

where $\omega = (\nabla_{2D}\times\mathbf{v})\cdot\hat{\mathbf{z}}$ is the out-of-plane vorticity field and $B_r(\mathbf{x})$ is a square of side $r$ centred at $\mathbf{x}$. The convolution kernel can be written as the product of two rectangular functions, $H_r(\mathbf{x}) = \Pi(x/r)\,\Pi(y/r)$, where $\Pi(x) = 1$ for $|x| < \frac{1}{2}$ and 0 otherwise. Note that we have used Stokes' theorem to recast the contour integral (1) as a surface integral of vorticity. The convolution in Eq. (12) can be efficiently computed in Fourier space using the Fourier transform of the rectangular kernel, which may be written in terms of the normalised sinc function as $\hat{H}_r(k_x, k_y) = (r/L)^2 \mathrm{sinc}(k_x r/2\pi)\,\mathrm{sinc}(k_y r/2\pi)$.

As mentioned earlier, the GP velocity field diverges at vortex locations. To minimise the numerical errors resulting from such singularities, we first resample each two-dimensional cut of the GP wave function field $\psi(\mathbf{x})$ into a very fine grid of resolution $32768^2$, using Fourier interpolation. The velocity field is then evaluated in physical space using the Madelung transformation. This resampling procedure is described in more detail in Ref. [13].

In NS simulations, the above algorithm is also applied to compute the coarse-grained dissipation $\varepsilon_r(\mathbf{x})$ over squares of side $r$. Instead of the vorticity, the convoluted quantity is in this case the dissipation field $\varepsilon(\mathbf{x}) = 2\nu s_{ij}s_{ij}$, where $\mathbf{s}(\mathbf{x}) = [\nabla\mathbf{v} + (\nabla\mathbf{v})^T]/2$ is the three-dimensional strain-rate tensor.

### Vortex detection from GP simulations

For a given two-dimensional cut of a GP velocity field, we identify the signs and locations of the quantum vortices crossing the cut as follows. First, the circulation is computed on a discrete grid following the procedure described above, taking small square loops of side $r \sim \xi \ll \ell$. The result is a discrete circulation field, where each circulation value is either zero if no vortex crosses the small loop centred at that position, or $\pm\kappa$ if a single vortex crosses it. For very small loop sizes, the former case is much more likely than the latter. As a result, the vortex distribution can be sparsely described by storing the locations and signs of the non-zero circulation values. By repeating this procedure over different cuts of the simulation, one can reconstruct the three-dimensional vortex structure, as visualised in Fig. 1.

### Energy spectrum computation from discrete vortices

For each two-dimensional cut, once the positions $\mathbf{r}_i$ and the signs $s_i$ of each vortex crossing the plane are determined, we first compute a regularised two-dimensional vorticity field $\omega(\mathbf{r}) = \kappa\sum_{i=1}^N s_i\delta_\eta(\mathbf{r}-\mathbf{r}_i)$, where $N$ is the number of vortices on the 2D cut. Here, $\delta_\eta(\mathbf{r}) = \exp(-|\mathbf{r}|^2/2\eta^2)/2\pi\eta^2$, and $\eta$ is the scale of the regularisation (we have used $\eta = \xi$ in Fig. 2). Then, the energy spectra are computed by noting that $|\hat{\mathbf{v}}(\mathbf{k})|^2 = |\hat{\omega}(\mathbf{k})|^2/|\mathbf{k}|^2$, where $\hat{\mathbf{v}}$ and $\hat{\omega}$ are the Fourier transforms at the wavevector $\mathbf{k}$ of the velocity field and of $\omega$, respectively. Finally, by averaging over all 2D cuts and integrating over a shell $|\mathbf{k}| = k$, the energy spectrum reads

$$E(k) = \frac{\kappa^2 |\hat{\delta}_\eta(k)|^2}{2k}\int\left\langle\sum_{i,j} s_i s_j e^{i\mathbf{k}\cdot(\mathbf{r}_i-\mathbf{r}_j)}\right\rangle\mathrm{d}\Omega, \quad (13)$$

where the integral is performed over all angles $\Omega$. Note that the large-wavenumber range in Fig. 2 is determined by the regularised Dirac function $\delta_\eta$ and has no physical meaning.

For disordered turbulence, as there is no correlation between the signs and the vortex positions, it is easy to show that $\left\langle\sum_{i,j} s_i s_j e^{i\mathbf{k}\cdot(\mathbf{r}_i-\mathbf{r}_j)}\right\rangle = \langle N\rangle$, from where it follows $E(k) \sim k^{-1}$.

## Data availability

Processed data used in the other figures are available from the corresponding authors upon request. Source data are provided with this paper.

## Code availability

Code used to process solution fields from GP and NS simulations is openly available at https://github.com/jipolanco/Circulation.jl and on Zenodo[65], along with detailed installation instructions and a complete set of examples. The software is licensed under the open-source Mozilla Public License 2.0.

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

## Acknowledgements

We acknowledge useful scientific discussions with L. Galantucci and S. Thalabard. This work was supported by the Agence Nationale de la Recherche through the project GIANTE ANR-18-CE30-0020-01. G.K. was also supported by the Simons Foundation Collaboration grant "Wave Turbulence" (Award ID 651471). This work was granted access to the HPC resources of CINES, IDRIS and TGCC under the allocation 2019-A0072A11003 made by GENCI. Computations were also carried out at the Mésocentre SIGAMM hosted at the Observatoire de la Côte d'Azur.

## Author contributions

Navier–Stokes and Gross–Pitaevskii simulations were performed by J.I.P. and N.P.M., respectively. J.I.P. and N.P.M. post-processed data. J.I.P., N.P.M and G.K. equally contributed to theoretical developments and writing the paper.

## Competing interests

The authors declare no competing interests.
