## [Peer Review File. · Nature Communications]

REVIEWER COMMENTS

Reviewer #1 (Remarks to the Author):

The manuscript presents results that may be valuable for readers familiar with turbulence modelling.

I especially like the idea of linking 'the spatial distribution of vortices in quantum turbulence to the coarse-grained energy dissipation in classical turbulence' (lines 7 and 8 of the abstract) and the resulting figure 5. The absence of distribution tails for low values of energy dissipation, loosely corresponding to flow regions with low vortex concentration, is specifically remarkable, at least in my view. Additionally, the vortex dipole discussion on page 6 is quite neat -- I like this too.

In essence, the authors describe the phenomenon of turbulence intermittency from a novel perspective. I therefore believe that the obtained results deserve to be published in some form.

On the other hand, the reported results are obtained for specific flows, mentioned solely in the methods section; that is, the results' relevance for actual flows -- which can be experimentally probed, I mean -- should be stated by the authors in more detail at some point of the main text.

On top of this, one needs the vorticity to get the circulation (see again the methods section); that is, one needs to know the fluid velocity and its derivatives to calculate the circulation and this is definitely not straightforward for most flows that can be experimentally probed, at least if one has in mind a quantitative study using state-of-the-art equipment.

I am pretty sure that the authors are aware of these issues. I am just saying that it would be nice to mention them, in order to clarify the conditions in which the proposed intermittency description was tested and their relevance to actual flows observed in nature.

Similarly, from lines 21 to 24, it is written that 'In three-dimensional flows ... energy is transferred from large to small scales through a cascade-like process'. Well, this is the case only for some flows, because, for example, energy can also be injected at small scales.

More generally, the authors sometimes attribute to any flow features that are instead observed only in some conditions. For example, references 6, 15, 16 and 29 report experimental results obtained in the two-fluid regime of superfluid helium-4, which cannot be quantitatively modelled within the GP framework used by the authors. Again, I believe that the issue should be mentioned, i.e. the conditions associated to the proposed intermittency description should be clearly identified and critically related to the existing literature.

To this end, I also want to mention that the relation between reconnections and irreversibility -- mentioned on lines 67 and 68 -- was already suggested, on the basis of experimental results, by Svancara and La Mantia, *J. Fluid Mech.* 876, R2 (2019).

Analogously, the occurrence of intermittency in quantum turbulence and the similarity of the latter with that observed in classical turbulence are apparent from experimental results, reporting the statistical distributions of velocity increments, characterized by classical-like tails at sufficiently small flow scales, see, for example, La Mantia and Skrbek, *Phys. Rev. B* 90, 014519 (2014) or Svancara and La Mantia, *J. Fluid Mech.* 832, 578 (2017).

On the other hand, the just mentioned papers report experimental results obtained in the two-fluid regime of superfluid helium-4, which, as already noted, is characterized by features absent within the GP framework, such as the fluid viscosity. Once more, I believe that the authors should clearly identify the conditions associated to the proposed intermittency description and critically related them to the existing literature.

The authors could also mention that 'the link between vortex polarisation and K41 statistics' (lines 182 and 183) has actually an experimental origin, discussed, for example, by Roche and Barenghi 2008 *EPL* 81, 36002 (2008).

I also want to note that the figures look a bit too busy to me and that the chosen colours are often too similar to each other, e.g. in figures 2 and 4. Additionally, it is mentioned that 'quasi-two-dimensional structures can be identified by eye in the inset of Fig. 1' (lines 455 and 456). Well, the authors could possibly indicate explicitly in the figure (some examples of) these structures.

Reviewer #2 (Remarks to the Author):

Please check the attached file.

Reviewer #3 (Remarks to the Author):

Refereed paper entitled "Vortex clustering, polarisation and circulation intermittency in classical and quantum turbulence" continues to study the problem of circulation intermittency presented in the recent paper "Intermittency of Velocity Circulation in Quantum Turbulence" of the same authors,

published in PRX, v 11, 011053 (2021) (Ref[12]). Both papers are based on their comprehensive 2048^3 direct numerical simulations of the generalized Gross–Pitaevskii (GP) equation for the quantum turbulence and incompressible Navier-Stokes (NS) equation for the classical turbulence. The paper contains some new and interesting results and new methodological approaches. However, many of the statements poised as new results or insights are in fact not new and sometimes presented in a somewhat biased way. I elaborate on some of these points below.

I should say that even the abstract immediately rises a few comments:

1) The statement "as no theory exists to explain their observed Spatio-temporal intermittency" is too strong. It ignores numerous studies of various aspects of intermittency, going back to the celebrated Kolmogorov 1962 paper (Ref [25]), mentioned in the refereed paper as K62-theory. Some of them one can find in the book by U. Frisch, Turbulence: The Legacy of A.N. Kolmogorov (Ref. [2]). A set of important results in the NS-based theory of intermittency one can find in papers by K. P. Zybin and V. A. Sirota, by E. Gkioulekas, and by many others. It is true that there is no final theory of intermittency like there is no final theory of quantum electrodynamics which finds the values of the electron charge and its mass. But the origin of intermittency and many of their properties are already clear now.

2) The statement "Turbulent flows may be regarded as a disordered collection of interacting vortices" is not accurate: there is some order in the statistics of vortices, like long-distant correlation in their orientations.

Also, starting from the title, abstract, and in the body of the paper, the authors used the name "vortex polarization", having in mind correlations of the vortex orientations, separated by some distance. This is not a good idea. I rather associate the term "vortex polarization" with their preferable orientation, which is definitely absent in isotropic turbulence.

As seen from the comparison of the titles of the refereed paper and Ref[12], the current paper is focused on vortex clustering and polarisation in the intermittency context. Indeed, as written in the Abstract "We show that, in quantum flows, Kolmogorov

turbulence emerges from vortex polarisation, while deviations – associated with intermittency – originate from their non-trivial spatial arrangement." The same statements on found in the body of the paper, in lines 269-273 and 506-510:

269 The previous results suggest that the fluctuations of
270 the spatial distribution of vortices are responsible for
271 the intermittency of the system, while the non-trivial
272 polarisation of vortices is the one that yields Kolmogorov

273 scalings.

506 we have shown that, in discrete-vortex systems, the Kolmogorov

507 self-similar scalings result from a partial polarisation

508 of the vortices, while the intermittency of circulation

509 statistics is linked to the non-trivial (non-Poissonian) spatial

510 distribution of vortices.

I should say here that both statements are not new and known for decades. For example, the authors themselves wrote, "Note that, in quantum turbulence, the link between vortex polarisation and K41 statistics was already suggested in previous works [21], and studied numerically in the context of the vortex filament model [22]". As for the second statement (non-trivial spatial distribution of vortices), this is commonplace in any multifractal models of intermittency.

What is new, at least for me, is an elegant method to illustrate these statements by randomization of the vortex directions in their tangle, leading to the so-called disordered quantum turbulence state. On the other hand, it would be very instructive here to compare the energy distribution over scales (the energy spectra) of the original GP-turbulence and the energy spectra of the disordered quantum turbulence state. Moreover, in order to understand better all author results (plots with the scaling exponents, etc.) one needs to see basic information about classical and quantum turbulence-comparison of their energy spectra, obtained from the GP- and NS- simulations. Unfortunately, I have not found this comparison either in the refereed paper or in their previous paper [12]. To present the energy spectra is all the more necessary because NS-turbulence is incompressible, while the GP equation describes the compressible flow. One has to see the answers to the following questions: What is the portion of incompressible flow energy with the comparison of that of compressible (sound) and potential flow? What is the typical level of non-linearity in their simulations? I strongly suggest including this information in the revised version of the paper.

One more author proposition as new elements in the refereed paper, which I'd like to comment, is formulated in the Discussion in the following lines:

520 Finally, using data from NS and GP simulations, we

521 have shown that the classical K62 theory does not fully

522 account for the intermittency of the circulation in classical

523 and quantum turbulence.

This is true. The only point is that classical K62 theory does not AIM

to account for the intermittency of the circulation.

523 We have provided an explanation
524 based on the presumed topology of the turbulent
525 structures that most contribute to extreme circulation
526 events. We have then proposed a modified K62 description
527 of circulation, where relevant singular structures have
528 a fractal dimension $D^\infty \approx 2.2$ associated with vortex sheets.
529 This value differs from the dimensionality $D^\infty = 1$ of
530 isolated vortex filaments, used in the modeling of velocity
531 increment statistics.

This explanation has been done before in paper by K.P.Iyer, K.R.Sreenivasan, P.K.Yeung, Phys.Rev.X, v 9,041006(2019), where it is written:

Regarding the result that $D=2.2$ for higher moments, one may infer that they are due to moderately wrinkled vortex sheets rather than more complex singularities.

531 Using this idea, we show
532 that the intermittency of circulation is well reproduced
533 by a modified version of the She–L'ev'equ model.

Actually, in the above-sited paper it was suggested a fit formula for the circulation scaling exponents $\lambda_p = h p + (3-D)$ which is the monofractal simplification with dimension $D=2.2$ of the more general multifractal She–L'ev'equ model. Therefore I cannot consider the statement in lines 531-533 as completely new.

534 All the previous ideas were additionally tested by introducing
535 a disordered quantum turbulent state, obtained by
536 artificially suppressing vortex polarisation from a GP
537 numerical simulation.

This is really new and interesting. I also consider a simple discrete model of circulation as new and interesting.

To characterize the refereed paper as a whole I did not find sufficiently impressive new results that are interesting for a wide physical community and therefore I can not recommend publishing the paper in Nature Communications.

On the other hand, I should say that the paper is well written, with a clear introduction and it includes a few rather interesting results. Also, the paper contains a set of methodological findings allowing a description of already known basic results in the theory of intermittency from another viewpoint, however without a clear statement than these results are not new. I think that the refereed paper can serve as a good basis for really important and interesting for the experts in hydrodynamics paper, which can be reconsidered for publication in specialized journals. Clearly, before that the text should be essentially revised, presumably accounting for my comments and criticisms. I hope that after revision the author's attempts to pull the blanket over themselves will be less obvious.

Reviewer #4 (Remarks to the Author):

I read with great interest the work by Polanco et al on circulation intermittency mainly because of how much I had enjoyed their earlier study on a similar theme which was published in Phys. Rev. X.

The paper as it stands is very good but in my opinion only an addition to the results reported in PRX. In particular, the big bang idea of calculating the exponents from the circulation and having a new way to look at what's intermittent in quantum turbulence was already done in the PRX work.

So the next question is if the claims in the abstract and the introduction are borne out by the additional

results and reinterpretation in terms of the dissipation field etc. Here I am not so convinced for the following reasons.

The introductory paragraph makes a sudden transition from vortices to the energy cascade a la Richardson. Now

in the Richardson (and modifications of it), the idea of vortices is neither implicit or explicit. This is equally

true of all the phenomenological models of classical turbulence (beta model, multifractal model, etc). So when

the authors try to recast their problem of vortices being the central story in the intermittency problem of

classical turbulence (it may well be) they mix the two approaches in my view. As a result, the final conclusions drawn,

which is certainly elegant and nice, is perhaps not as major a breakthrough as the authors envisage it to be. Hence statements

about "the existence of a coarse-grained quantity different from ϵ_r " to capture intermittency while being perhaps true

is not as strong a result, in my opinion, to merit publication in Nature Comm. The analogy between classical and quantum

turbulence in terms of circulation is certainly clear and compelling but I think the PRX paper already makes this case.

The new addition of then using this to understand intermittency in classical turbulence is not, in my opinion, as solid.

Specifically,

1. The adaptation of K62/She-Leveque/dissipation-multifractal ideas while being nice stops short of saying

where exactly the nature of vortices (in the light of the authors' work) comes in. I know one can reinterpret

the "dimensional" parameter D in such models in the spirit of dimensions of the vortical fields but this is

very different from the sharper definition of quantized vortices in quantum turbulence.

2. The inhomogeneity of the spatial distribution of vortices in the quantum problem is an important input in

this story and its understanding by contrasting the toy model and the GP simulations is an excellent step even

if its not for understandable reasons totally rigorous. For classical turbulence, identifying such "filaments" is not easy.

Often people resort to QR plots etc as a result. Therefore here again the connections between classical and quantum turbulence

is far from straightforward to settle the questions in a definite manner.

Reply to Reviewer #1

We thank the Referee for their review and positive comments, and for supporting the publication of our manuscript in Nature Communications. In the following, we provide a point-by-point response for the remarks raised in the report. Quotes from the Referee’s review are in bold fonts, while relevant modifications to the manuscript are in blue.

Remark 1

The manuscript presents results that may be valuable for readers familiar with turbulence modelling.

I especially like the idea of linking ‘**the spatial distribution of vortices in quantum turbulence to the coarse-grained energy dissipation in classical turbulence**’ (lines 7 and 8 of the abstract) and the resulting figure 5. **The absence of distribution tails for low values of energy dissipation, loosely corresponding to flow regions with low vortex concentration, is specifically remarkable, at least in my view. Additionally, the vortex dipole discussion on page 6 is quite neat – I like this too.**

In essence, the authors describe the phenomenon of turbulence intermittency from a novel perspective. I therefore believe that the obtained results deserve to be published in some form.

We are glad to hear that the Referee appreciates our work and recommends the manuscript for publication.

Remark 2

On the other hand, the reported results are obtained for specific flows, mentioned solely in the methods section; that is, the results’ relevance for actual flows – which can be experimentally probed, I mean – should be stated by the authors in more detail at some point of the main text.

We thank the Referee for this remark. We agree that a discussion about circulation in more general models and the possibility of experimental confirmation was indeed missing. We added a paragraph to the Discussion section discussing the robustness of our results and their potential experimental realisation:

It is important to remark that the quantum turbulence simulations presented in this work have been performed on periodic domains, and are based on the GP equation describing an ideal superfluid at very low temperature. In contrast, most superfluid turbulence experiments using liquid helium are performed in confined systems and at finite temperatures^{18,19}, in a regime that may be described by a two-fluid model⁵¹. Early experimental studies showed that the signature of intermittency on velocity increments is nearly independent of the temperature, matching observations in classical fluids^{52–54}. These observations were later contradicted by a recent experimental investigation, which showed an enhancement of velocity intermittency in the two-fluid regime compared to classical turbulence⁵⁵, in agreement with previous numerical simulations of related models^{56,57}. Compared to velocity increments, we expect the circulation to be a much more robust observable in quantum fluids, as it does not display singular behaviour in the vicinity of

vortices¹⁴. For this reason, measuring the scaling properties of circulation in future experiments may help disambiguate existent contradictions, and provide a clearer answer on the intermittency of finite-temperature quantum turbulence. Recent experiments have succeeded to reconstruct Eulerian velocity fields from Lagrangian particle tracking measurements in turbulent superfluid helium⁵⁵. Such a technique could be used in principle to measure the velocity circulation in superfluid helium, although addressing high-order statistics might still be challenging.

Remark 3

On top of this, one needs the vorticity to get the circulation (see again the methods section); that is, one needs to know the fluid velocity and its derivatives to calculate the circulation and this is definitely not straightforward for most flows that can be experimentally probed, at least if one has in mind a quantitative study using state-of-the-art equipment.

In this work, we indeed obtained the circulation from the vorticity field, as detailed in the Methods section of our manuscript. However, it is also possible to obtain it directly from the velocity field, thus avoiding the need for spatial derivatives. This was done both in Iyer *et al.*, *Phys. Rev. X* 2019 and in Müller *et al.* *Phys. Rev. X* 2021. Numerically, we have verified that both approaches give the exact same result up to numerical precision.

Experimentally, one would need to be able to obtain instantaneous Eulerian velocity fields, noting that a two-component two-dimensional cut is sufficient. In the past, this has been indeed done using PIV in classical turbulence experiments (see e.g. Sreenivasan *et al.* *PRL* 1995, Zhou *et al.* *JFM* 2008). We realise that this approach is more challenging in superfluid helium, which is why an alternative approach that may be worth considering would be to reconstruct Eulerian velocity fields from Lagrangian particle tracking measurements. We have added a comment in the main text regarding this issue (see Remark 2).

Remark 4

Similarly, from lines 21 to 24, it is written that ‘In three-dimensional flows ... energy is transferred from large to small scales through a cascade-like process’. Well, this is the case only for some flows, because, for example, energy can also be injected at small scales.

We clarified that this scenario takes place when energy is injected at large scales:

In three-dimensional flows, because of the inherently non-linear character of turbulence, **energy initially injected at large scales is transferred towards the small scales** through a cascade-like process.

Remark 5

More generally, the authors sometimes attribute to any flow features that are instead observed only in some conditions. For example, references 6, 15, 16 and 29 report experimental results obtained in the two-fluid regime of superfluid helium-4, which cannot be quantitatively modelled within the GP framework used by the authors. Again, I believe that the issue should be mentioned, i.e. the conditions associated to the proposed intermittency description should be clearly identified and critically related to the existing literature.

In the new paragraph quoted in Remark 2, we included a discussion on the links between superfluid helium-4 experiments and the numerical simulations we perform. We also included a discussion about intermittency of velocity increments in superfluids, as well as motivating the study of the intermittency of circulation in finite-temperature superfluid helium.

Remark 6

To this end, I also want to mention that the relation between reconnections and irreversibility – mentioned on lines 67 and 68 – was already suggested, on the basis of experimental results, by Svancara and La Mantia, *J. Fluid Mech.* **876**, R2 (2019).

We thank the Referee for their remark. We have included this reference in the text.

Remark 7

Analogously, the occurrence of intermittency in quantum turbulence and the similarity of the latter with that observed in classical turbulence are apparent from experimental results, reporting the statistical distributions of velocity increments, characterized by classical-like tails at sufficiently small flow scales, see, for example, La Mantia and Skrbek, *Phys. Rev. B* **90**, 014519 (2014) or Svancara and La Mantia, *J. Fluid Mech.* **832**, 578 (2017).

We thank the Referee for these references on the distributions of Lagrangian velocity increments in superfluid helium. We now cite both references in the text.

Remark 8

On the other hand, the just mentioned papers report experimental results obtained in the two-fluid regime of superfluid helium-4, which, as already noted, is characterized by features absent within the GP framework, such as the fluid viscosity. Once more, I believe that the authors should clearly identify the conditions associated to the proposed intermittency description and critically related them to the existing literature.

Please see the answer to Remark 2, and in particular the new paragraph added to the text.

Remark 9

The authors could also mention that ‘the link between vortex polarisation and K41 statistics’ (lines 182 and 183) has actually an experimental origin, discussed, for example, by Roche and Barenghi 2008 *EPL* **81**, 36002 (2008).

We thank the Referee for pointing out that reference. We included this reference in the text where we discuss previous works linking polarisation and K41 (lines 91-93 and 224-228 of the revised version).

Remark 10

I also want to note that the figures look a bit too busy to me and that the chosen colours are often too similar to each other, e.g. in figures 2 and 4. Additionally, it is mentioned that ‘quasi-two-dimensional structures can be identified by eye in the inset of Fig. 1’ (lines 455 and 456). Well, the authors could possibly indicate explicitly in the figure (some examples of) these structures.

We thank the Referee for their comments on the figures. We removed the lines saying that these structures can be identified by eye because this may introduce some subjectivity, and we have not managed to create a better visualisation of this.

We also performed a few changes on the figures to make them look less busy:

- Old figure 2 (now 3): We moved the insets in panel (a) to individual panels (b) and (c).
- Old figure 4 (now 5): We added an arrow to make it easier to the eye to understand the colour convention. We also removed some curves that made the figure look busy, namely the vertical lines in

panel (a) and some Poisson distributions in panel (b). We also modified the colour of some curves for a better visualisation.

Reply to Reviewer #2

We thank the Referee for their review and positive comments, and for supporting the publication of our manuscript in Nature Communications. In the following, we provide a point-by-point response for the remarks raised in the report. Quotes from the Referee’s review are in bold fonts, while relevant modifications to the manuscript are in green.

Remark 1

In my view, the paper has a strong potential to trigger considerable attention from the turbulence community and more general readers, in consonance with the scientific standards of Nature Communications.

We are glad to see that the Referee appreciates our work and finds it suitable for Nature Communications.

Remark 2

What is the difference between the averaging procedure taken for $\langle n^{\gamma_p} \rangle_\tau$ in Eq. (4) and the one taken for $\langle n^p \rangle_A$ in Eq. (6)? The authors have used a different notation for these averages, but they seem to represent identical procedures. In this case, using the relation between n and ϵ_r , established as Eq. (5), one would get, from Eq. (4), the circulation scaling exponent

$$\lambda_p = 4p/3 + \tau(2p/3),$$

which strongly disagrees with data, for any reasonable choice of the energy transfer intermittency exponent $\tau(2p/3)$. The authors should clarify this issue.

We thank the Referee for their very assertive comment. Indeed, in the urge to try to motivate the relation between the scaling exponents in a handwaving manner, we introduced some relations making use of conditioned statistics. The reasoning done in Eq. (4) does not arise from an exact computation and overlooks some conditionings that have to be done, which are actually difficult, maybe impossible, to handle theoretically. To avoid confusions, and remain purely speculative, we have modified Eq. (4) to simply

$$\lambda_p = g(\gamma_p)$$

Remark 3

We thank the Referee for their constructive remarks concerning the modelling of the circulation exponents. We are amazed by how this new and different approach can also accurately predict the signature of intermittency on the exponents.

Very interestingly, in the model proposed by the Referee, if ones fixes the exponent α following their suggestion (i.e. such that $\lambda_3(\alpha) = 4$) one finds $\alpha = -0.059$, quite different from the traditional value of -0.03 , but perhaps closer to the value -0.053 recently obtained in high Reynolds numbers Navier–Stokes simulations Iyer, Sreenivasan & Yeung, *Phys. Rev. Fluids* 5, 054605 (2020). We completely agree with the Referee’s view

that assuming $\lambda_3 = 4$ is nothing but an educated guess as there is no exact result fixing this exponent. Therefore, such a finding should not be pushed further.

Before suggesting their new model, the Referee states:

The inviscid K41-like relation $\Gamma \sim \epsilon_r^{1/3} r^{4/3}$ is not, actually, the correct way to address the statistics of circulation in classical turbulence, a fact that has been the source of some confusion in the literature.

We believe that there is not yet a sharp view on this issue, as indeed different works assume different scalings of Γ with the local dissipation rate. The relation $\Gamma \sim \epsilon_r^{1/3} r^{4/3}$ is motivated by the fact that the circulation can be expressed in terms of the transverse velocity structure functions. This was noticed, for instance, by Iyer et al., *Phys. Rev. X* 9, 041006, (2019), and before this by Cao, Shen & Sreenivasan, *Phys. Rev. Lett* 76, 616 (1996). Indeed, if we consider a square loop with vertices $(0, 0)$, $(0, r)$, $(r, 0)$, (r, r) , the circulation around it can be directly written as

$$\Gamma = \int_0^r (v_x(x, 0) - v_x(x, r))dx - \int_0^r (v_y(0, y) - v_y(r, y))dy = \int_0^r \delta_r^y v_\perp(x, 0)dx - \int_0^r \delta_r^x v_\perp(0, y)dy,$$

where $\delta_r^{\{x, y\}} v_\perp(x, y)$ is the transverse velocity increment at point (x, y) along the $\{x, y\}$ direction. From this perspective, one could expect Γ to be rather related to the fluctuations of $\delta_r v_\perp$, and therefore to $\epsilon_r^{1/3}$. However, as stated in our work, because the circulation is the surface integral of the vorticity, there are likely important vorticity cancellations that should be, at least in some phenomenological manner, taken into account. This is exactly our motivation to consider a value of $D_\infty \neq 1$, as it is explained in the text. We recognise however that, as all multifractal models, our modified Obukhov–Kolmogorov OK62 model is based on purely phenomenological assumptions.

Instead of the OK62-based relation $\Gamma \sim \epsilon_r^{1/3} r^{4/3}$, the Referee proposes:

... one may write, for a disk of radius r

$$\Gamma_r \sim \omega_r r^2$$

where $\omega_r \sim \sqrt{\epsilon_r/\nu_r}$ is a scale-dependent vorticity fluctuation defined in terms of the scale-dependent energy transfer rate ϵ_r and the Kraichnan’s eddy viscosity. It follows that

$$\Gamma \sim \omega_r r^2 \sim \epsilon_r^{1/2} \nu_r^{-1/2} r^2$$

We totally agree with the Referee that, if ones considers fluctuations of velocity gradients ∂v (an inherently small-scale quantity), one should indeed have $\partial v \sim \epsilon^{1/2}$, as done for instance in Wyngaard & Tennekes, *Phys. Fluids* 13, 1962 (1970) and Kholmyansky, Moriconi, Pereira & Tsinober, *Phys. Rev. E* 80, 036311 (2009). However, the extension of such a scaling to the coarse-grained vorticity at a scale r in the inertial range is far from obvious, and the use of an eddy-viscosity is purely phenomenological. In addition, obtaining the circulation scaling exponents from such modelling requires a series of mean-field approximations that, although perhaps intuitive, remain phenomenological. Indeed, the determination of the intermittency exponents by following the Referee’s ideas rests on the (traditional) idea that the eddy-viscosity is not a fluctuating quantity, but is determined through the energy spectrum. The latter is however a two-point quantity that depends on the mean energy dissipation and on the intermittency correction α . By definition, the eddy-viscosity is thus an average quantity that depends only on second order statistics, which is then used, together with ϵ_r , to predict the behaviour of high-order moments. We find the Referee’s ideas very interesting but far from being a rigorous theory.

The series of arguments and ideas proposed by the Referee indeed open the possibility for a different interpretation of circulation intermittency that is worth to be discussed and to be developed further. However, we believe that including the full Referee’s derivation in the manuscript will deviate the attention of the reader from the main messages we are trying to deliver. We judge nevertheless important to make the reader aware of the different approach suggested by the Referee. Therefore, we have modified the last paragraph before the Discussion section:

The previous results provide a possible interpretation for the difference between the intermittency of velocity fluctuations and of circulation, based on the different topologies of the dissipative structures contributing to extreme events. We shall notice that, as suggested by an anonymous Referee, an alternative interpretation is also possible, based on the recent works by Apolinário *et al.*³⁴ and Moriconi³⁵. In this framework, the circulation should scale as $\varepsilon_r^{1/2}$ (instead of $\varepsilon_r^{1/3}$), namely $\Gamma \sim \varepsilon_r^{1/2} \nu_r^{-1/2} r^2$, where ν_r is Kraichnan’s eddy viscosity⁵⁰. The latter is found by assuming that the energy spectrum takes the form $E(k) \sim \varepsilon^{2/3} k^{-5/3+\alpha}$ (where α is an intermittency correction), yielding $\nu_r \sim r^{4/3+\alpha}$. Note that this phenomenological approach mixes a mean-field approximation for determining ν_r with the fluctuations arising from $\varepsilon_r^{1/2}$. It also obscures the possibility of vortex cancellations. Nevertheless, when combined with the standard She–Lévêque model (with $D_\infty = 1$), this model provides an expression for the exponents λ_p as accurate as our mOK62 model in the turbulent case. There is certainly a need to pursue further investigations to understand how both models differ and complement each other.

We hope that the Referee understands our decision of keeping our model and narrative, and that they find fair the new added discussion about their modelling. We kindly invite the Referee, if they wish, to contact us after revisions to discuss further about these interesting issues that could trigger a fruitful collaboration.

Remark 4

From the previous discussion, it should be clear that it is not necessary, as addressed by the authors, to claim that two-dimensional singular structures would be essential to model circulation scaling exponents. The standard She–Lévêque model with $D_\infty = 1$ suffices. This, in my view, would render the paper much more attractive, once it would unify the discussion of the scaling exponents λ_p and $\alpha(p)$, Eq. (6), within the framework of the uniquely defined She–Lévêque intermittency exponents $\tau(q, D_\infty = 1)$.

We thank the Referee for this comment. Indeed, in the answer of Remark 3, we mention that this alternative approach correctly works with the standard dimensionality $D_\infty = 1$ of the most singular structures appearing in the She–Lévêque model.

I add that the authors’ attempt to get support for the “ $D_\infty = 2$ theory”, from the idea that “Such quasi-two-dimensional structures can be indeed identified by eye in the inset of Fig. 1.” (Lines 455-456) besides being very subjective, sounds excessively self-assertive, likely to find hard consensus among readers.

We have removed the lines in the text where we mentioned that these structures can be identified in Fig. 1, as this can indeed be subjective.

I guess, furthermore, that the phenomenological puzzle pointed out in the above item (1) could be revisited now under a deeper perspective, since the contribution of eddy viscosity scaling, relevant for the evaluation of circulation moments, does not come into play in the derivation of Eq. (6).

Indeed, this is a very interesting topic that will be studied in a future work. We have included a comment in the Discussion:

An alternative approach, proposed by an anonymous Referee, suggests that the most relevant singular structures for velocity circulation should still be vortex filaments. Further investigations on the fractal dimension of circulation would help develop more accurate models of intermittency.

Remark 5

The use of the word “coefficient” in Lines 241-242, “These exponents are compared to the measured λ_p^{turb} coefficients (...)” is a bit uncommon. May be “These exponents are compared

to the measured values of λ_p^{turb} (...)” would sound more appropriate.

We thank the Referee for this assertive comment. We have changed the text with the suggestion they proposed.

Remark 6

The authors should provide references for the statements that start at Line 27 and end at Line 30 (about the thickness of classical vortex filaments), and for the ones that start at Line 77 and end at Line 80 (standard information about superfluid Helium).

We have added some references for these statements.

Remark 7

The authors make reference to the “Kolmogorov-Obukhov K62” theory of intermittency. Why not to use the acronym KO62? Alternatively, I would like to point, together with Frisch [Proc. Roy. Soc. Lond. 434, 89 (1991)], that for the sake of historical justice, one should refer, instead, to the “Obukhov-Kolmogorov” (then, OK62) theory of intermittency.

Again, we thank the Referee for this comment. We now refer in the text to this theory as OK62.

Reply to Reviewer #3

We thank the Referee for their review and their constructive comments. In the following, we provide a point-by-point response for the remarks raised in the report. Quotes from the Referee’s review are in bold fonts, while relevant modifications to the manuscript are in orange.

Remark 1

The paper contains some new and interesting results and new methodological approaches. However, many of the statements poised as new results or insights are in fact not new and sometimes presented in a somewhat biased way.

We are glad to hear that the Referee thinks that our work contains interesting results and new methodological approaches. We would like to apologise if our manuscript inclined the Referee to think that we are taking appropriation of known results in turbulence. We recognise and appreciate the effort made by the community working in turbulence. We carefully cited and discussed many relevant works related to turbulence, and stated clearly when some ideas were known by the classical or quantum turbulence communities. In the revised version we have included more references and state even more clearly if any idea was known by the community.

Remark 2

I should say that even the abstract immediately rises a few comments:

1) The statement “as no theory exists to explain their observed Spatio-temporal intermittency” is too strong. It ignores numerous studies of various aspects of intermittency, going back to the celebrated Kolmogorov 1962 paper (Ref [25]), mentioned in the refereed paper as K62-theory. Some of them one can find in the book by U. Frisch, *Turbulence: The Legacy of A.N. Kolmogorov* (Ref. [2]). A set of important results in the NS-based theory of intermittency one can find in papers by K. P. Zybin and V. A. Sirota, by E. Gkioulekas, and by many others. It is true that there is no final theory of intermittency like there is no final theory of quantum electrodynamics which finds the values of the electron charge and its mass. But the origin of intermittency and many of their properties are already clear now.

We agree with the Referee’s remark, as we do not take lightly the wealth of existent works that have much advanced our knowledge on the intermittency issue. Our intention was only to state, very briefly, that there are still many open questions on this problem and there is no exact first-principles theory (derived from Navier-Stokes equations) able to explain turbulence. To clarify this issue, we now write in the abstract:

[...] as no **first-principles** theory exists [...]

We have also extended the paragraph including Eq. 2, citing works by Zybin & Sirota *Phys. Rev. E* 85, 056317 (2012) and *Phys. Rev. E* 88, 043017 (2013), and the book by Uriel Frisch.

2) The statement “Turbulent flows may be regarded as a disordered collection of interacting vortices” is not accurate: there is some order in the statistics of vortices, like long-distant correlation in their orientations.

By “disordered” we did not mean decorrelated. To avoid any possible confusion, we have modified that sentence to:

Turbulent flows may be regarded as an **intricate** collection of **mutually-interacting** vortices.

Remark 3

Also, starting from the title, abstract, and in the body of the paper, the authors used the name “vortex polarization”, having in mind correlations of the vortex orientations, separated by some distance. This is not a good idea. I rather associate the term “vortex polarization” with their preferable orientation, which is definitely absent in isotropic turbulence.

We understand the possible confusion that may be caused by the term “polarisation”. However, this precise term has already been used in the superfluid turbulence community to refer to the correlations of the vortex orientations, and it appears sensible for us to preserve their usage. Some previous works using this term with this precise meaning include Vinen & Niemela, *J. Low Temp. Phys.* 128, 167 (2002); L’vov, Nazarenko & Rudenko, *Phys. Rev. B* 76, 024520 (2007); Roche & Barenghi, *EPL* 81 36002 (2008); Baggaley, Laurie, & Barenghi, *Phys. Rev. Lett.* 109, 205304 (2012); Barenghi, L’vov & Roche, *PNAS* 111, 4683 (2014).

To avoid possible confusion, we have replaced “polarisation” in the abstract by the following:

We show that, in quantum flows, Kolmogorov turbulence emerges from the **correlation of vortex orientations**, while deviations [...]

Furthermore, in the introduction, we have added a new paragraph introducing this term and its meaning in the context of our work. In this new paragraph, we cite previous works that have also used this term in quantum turbulence. Finally, to avoid confusion, we disambiguate the usage of “polarisation” with the idea of a preferential large-scale vortex orientation:

Previous studies have suggested that, in quantum turbulence, the emergence of K41 scaling laws is associated to a local *polarisation* of the vortex tangle^{23–28}. In other words, within a given spatial region, the orientations of nearby vortices are not independent, but instead have some degree of correlation. This phenomenon is visible in Fig. 1, where vortex bundles – regions of same-coloured vortex filaments – can be clearly identified. This local polarisation is present even in ideally isotropic flows, and should not be confused with the preferential large-scale orientation of vortices, which typically occurs in anisotropic flows. A classical example of the latter is a rotating cylindrical vessel filled with superfluid helium⁵.

Remark 4

As seen from the comparison of the titles of the refereed paper and Ref[12], the current paper is focused on vortex clustering and polarisation in the intermittency context. Indeed, as written in the Abstract “We show that, in quantum flows, Kolmogorov turbulence emerges from vortex polarisation, while deviations – associated with intermittency – originate from their non-trivial spatial arrangement.” The same statements are found in the body of the paper, in lines 269-273 and 506-510:

The previous results suggest that the fluctuations of the spatial distribution of vortices are responsible for the intermittency of the system, while the non-trivial polarisation of vortices is the one that yields Kolmogorov scalings.

we have shown that, in discrete-vortex systems, the Kolmogorov self-similar scalings result from a partial polarisation of the vortices, while the intermittency of circulation statistics is linked to the non-trivial (non-Poissonian) spatial distribution of vortices.

I should say here that both statements are not new and known for decades. For example, the authors themselves wrote, “Note that, in quantum turbulence, the link between vortex polarisation and K41 statistics was already suggested in previous works [21], and studied numerically in the context of the vortex filament model [22]”. As for the second statement (non-trivial spatial distribution of vortices), this is commonplace in any multifractal models of intermittency.

We do not claim at all to be the first to relate polarisation with Kolmogorov turbulence, as we indeed gave credit citing some previous works. To make this point clearer, as detailed in the Remark 3, we have added a new paragraph to the introduction on previous works suggesting the link between polarisation and K41 statistics.

Moreover, to clarify one of the novelties of our approach (the suggestion that polarisation has no impact on intermittency), we modified the above-cited paragraph to read:

The previous results suggest that the non-trivial polarisation of vortices, while being responsible for Kolmogorov scalings, has no major influence on the intermittency of the system. Furthermore, they indicate that the latter originates from fluctuations of the spatial distributions of vortices.

Concerning multifractal models, to our knowledge, classical approaches such as the random β -model Benzi et al., J. Phys A (1984) and refined models, are based on the idea of the fractality of the Richardson *cascade*, as large eddies are split into smaller eddies that occupy some fraction of the spatial volume. While there is implicitly some notion of spatial distribution of eddies there, it is very different from the more precise notion of instantaneous spatial distribution of discrete vortices that we use in our work, where the idea of a cascade has not been invoked.

Remark 5

What is new, at least for me, is an elegant method to illustrate these statements by randomization of the vortex directions in their tangle, leading to the so-called disordered quantum turbulence state. On the other hand, it would be very instructive here to compare the energy distribution over scales (the energy spectra) of the original GP-turbulence and the energy spectra of the disordered quantum turbulence state. Moreover, in order to understand better all author results (plots with the scaling exponents, etc.) one needs to see basic information about classical and quantum turbulence- comparison of their energy spectra, obtained from the GP- and NS- simulations. Unfortunately, I have not found this comparison either in the refereed paper or in their previous paper [12]. To present the energy spectra is all the more necessary because NS-turbulence is incompressible, while the GP equation describes the compressible flow. One has to see the answers to the following questions: What is the portion of incompressible flow energy with the comparison of that of compressible (sound) and potential flow? What is the typical level of non-linearity in their simulations? I strongly suggest including this information in the revised version of the paper.

The Referee raises important questions concerning GP turbulence that have been largely covered by many studies since the first paper on the subject in 1997 Nore et al., Phys. Fluids 9, 2644 (1997). In short, once Kolmogorov turbulence is observed, the dynamics is dominated by vortices and the energy of compressible parts plays no major role. We have added a sentence about this matter in lines 168-172.

We had initially judged unnecessary to show the energy spectra of the flows as they are well known by the community of quantum turbulence and they are simple second-order statistics quantities. Concerning the disordered turbulent spectrum, which is less known, it displays a trivial k^{-1} scaling, as by definition all correlations cancel. However, we agree with the Referee that showing them will help the readers and will make the manuscript self-consistent.

We have added a new figure (Fig. 2) showing the energy spectra obtained from NS and GP simulations. Note that, in the GP case, we plot the incompressible part of the kinetic energy spectrum. Moreover, to compare

the “turbulent” and the “disordered” cases, we have included in the figure the spectra obtained from the discrete vortices detected from the GP fields.

We have added the following text describing the new figure right before the Results section:

To illustrate the differences between the turbulent (non-disordered) and the disordered turbulence states, we plot in Fig. 2 the kinetic energy spectrum associated to each vortex configuration (see Methods for details on the computation of the spectra from discrete vortices). First, we see that the turbulent case displays a clear $k^{-5/3}$ range, in agreement with the energy spectra obtained from the full GP and NS fields. Note that, in the case of GP fields, we show the *incompressible* kinetic energy spectrum, which contains 86% of the total energy of the system – the other components being the compressible, internal and quantum energy^{20,22}. Secondly, the K41 scaling disappears once polarisation is artificially suppressed from the tangle, leading to a trivial k^{-1} scaling range for the disordered state (see Methods for a brief derivation). Note that this same scaling has already been observed in vortex filament simulations, once the vortex tangle has been decomposed into polarised and random components²⁶.

Note that more details on the GP simulations, including the spectra associated to the incompressible and compressible parts of the kinetic energy, are given in the paper Müller & Krstulovic, *Phys. Rev. B* 102, 134513 (2020), which is also cited in our present manuscript.

Remark 6

One more author proposition as new elements in the refereed paper, which I’d like to comment, is formulated in the Discussion in the following lines:

Finally, using data from NS and GP simulations, we have shown that the classical K62 theory does not fully account for the intermittency of the circulation in classical and quantum turbulence.

This is true. The only point is that classical K62 theory does not AIM to account for the intermittency of the circulation.

We agree with the Referee, as we did not mean to imply that K62 was expected to account for the intermittency of the circulation.

To avoid misunderstanding, we have replaced “shown” by “confirmed” in the sentence cited by the Referee.

Also note that, towards the end of the Results section, we have a paragraph that is precisely titled “Can Obukhov–Kolmogorov’s OK62 theory describe circulation intermittency?” (page 7 of revised version), where we briefly give a negative answer to this question – in agreement with earlier observations that we cite. While there was no reason to expect a positive answer to it, we believe that it is worth commenting on this question at that point of the paper. Indeed, it comes at a point where we have just shown that the coarse-grained dissipation ε_r , at the core of OK62 theory, is statistically equivalent to the number of vortices n per unit area in quantum turbulence (Figs. 5 and 6 in revised version), which in our opinion is far from being a trivial result. Moreover, conditioning circulation statistics on ε_r (or n), as we do in Fig. 6, is precisely motivated by OK62 theory, where the same is done for velocity increment statistics. For this reason, we believe that a discussion on the pertinence of OK62 theory is perfectly appropriate, and even necessary, at that point of the paper.

Remark 7

We have provided an explanation based on the presumed topology of the turbulent structures that most contribute to extreme circulation events. We have then proposed a modified K62 description of circulation, where relevant singular structures have a fractal dimension $D_\infty \approx 2.2$ associated with vortex sheets. This value differs

from the dimensionality $D_\infty = 1$ of isolated vortex filaments, used in the modeling of velocity increment statistics.

This explanation has been done before in paper by K.P.Iyer, K.R.Sreenivasan, P.K.Yeung, Phys.Rev.X, v 9,041006(2019), where it is written: Regarding the result that $D = 2.2$ for higher moments, one may infer that they are due to moderately wrinkled vortex sheets rather than more complex singularities.

We thank the Referee for noticing that this interpretation was directly inspired by Iyer et al.'s work. Please note that, in the first version of the manuscript (lines 465-472), we explicitly cited that work when we first introduced the fractal dimension $D_\infty = 2.2$:

To check this idea, we adapt the She–Lévêque model $\tau_{\text{SL}}(p)$ [Eq. (7)] by setting $D_\infty = 2.2$ instead of 1. The chosen dimensionality exactly corresponds to the monofractal fit obtained by Iyer et al. [20] et al. and Müller et al. [12] for the high-order circulation moments ($p > 3$) in classical and quantum turbulence, and, as suggested in the former work, it may be linked to the effect of wrinkled vortex sheets.

We now cite this work again in the paragraph quoted by the Referee. Moreover, later in the same paragraph, we have added:

Using this idea, we have shown that the intermittency of circulation is well reproduced by a modified version of the She–Lévêque model, bringing new support to the vortex sheet interpretation first proposed by Iyer et al.²⁹.

Remark 8

Using this idea, we show that the intermittency of circulation is well reproduced by a modified version of the She–Lévêque model.

Actually, in the above-sited paper it was suggested a fit formula for the circulation scaling exponents $\lambda = hp + (3 - D)$ which is the monofractal simplification with dimension $D = 2.2$ of the more general multifractal She–Lévêque model. Therefore I cannot consider the statement in lines 531-533 as completely new.

We thank the Referee for noticing this relation between Iyer et al.'s monofractal fit and the She–Lévêque model.

We now comment this in the text:

The chosen dimensionality exactly corresponds to the monofractal fit obtained by Iyer et al.²⁹ and Müller et al.¹⁴ for the high-order circulation moments ($p > 3$) in classical and quantum turbulence, and, as suggested in the former work, it may be linked to the effect of wrinkled vortex sheets. Note that, for large \$p\$, our mOK62 model simplifies to \$\lambda_p \approx \frac{10}{9}p + (3 - D_\infty)\$, which is equivalent to the monofractal fit by Iyer et al.²⁹. In Fig. 4, it is shown that the adapted model matches strikingly well the anomalous exponents of circulation both in the turbulent and in the disordered cases for $p > 3$ (dashed lines), while for $p < 3$ there are some deviations.

Remark 9

All the previous ideas were additionally tested by introducing a disordered quantum turbulent state, obtained by artificially suppressing vortex polarisation from a GP numerical simulation.

This is really new and interesting. I also consider a simple discrete model of circulation as new and interesting.

We thank the Referee for their interest in these elements of our work.

Reply to Reviewer #4

We thank the Referee for their review and positive comments. In the following, we provide a point-by-point response for the remarks raised in the report. Quotes from the Referee's review are in bold fonts.

Remark 1

The paper as it stands is very good but in my opinion only an addition to the results reported in PRX. In particular, the big bang idea of calculating the exponents from the circulation and having a new way to look at whats intermittent in quantum turbulence was already done in the PRX work.

We regret if the objectives and main messages of our present work were not sufficiently clear for the Referee. In our PRX paper we indeed showed the equivalence of the circulation statistics in classical and quantum turbulence. In the present work, this statistical equivalence is the starting point for building a mathematical toy model, performing a novel type of data analysis and unveiling a connection between the source of intermittency of circulation in classical and in quantum turbulence.

More precisely, in this work we study sets of discrete vortices extracted from our quantum turbulence simulations, which is very different from what we did in our PRX paper (where only Eulerian fields were used). This allows us to have a precise estimation of the polarisation and of the spatial distribution of vortices, something that is otherwise much more ambiguous to define when using continuous fields – in particular in classical flows, where vortices are not discrete objects with fixed circulation. Moreover, it allows us to artificially suppress polarisation, which enables us to separate both the effects of polarisation and spatial distribution. Then, we provide a deeper link between quantum and classical flows, we establish an equivalence between the spatial vortex distribution in quantum turbulence and the coarse-grained dissipation in classical turbulence, backed by numerical simulations of both systems. This equivalence justifies the application of results from discrete-vortex systems (i.e. quantum turbulence) to the classical case, which is what we do in the last part of the paper.

In addition, this new work provides a clearer view of intermittency in quantum turbulence, as we can now associate it to the inhomogeneous spatial distribution of vortices. This conclusion was not present in (and was not in the scope of) our PRX work. We are convinced that the results of the present work cannot be judged as incremental.

Remark 2

So the next question is if the claims in the abstract and the introduction are borne out by the additional results and reinterpretation in terms of the dissipation field etc. Here I am not so convinced for the following reasons.

The introductory paragraph makes a sudden transition from vortices to the energy cascade a la Richardson. Now in the Richardson (and modifications of it), the idea of vortices is neither implicit or explicit. This is equally true of all the phenomenological models of classical turbulence (beta model, multifractal model, etc). So when the authors try to recast their problem of vortices being the central story in the intermittency problem of classical turbulence (it may well be) they mix the two approaches in my view.

We agree with the Referee that the discussion of the Richardson cascade in the introduction may lead to some confusion, as our approach does not rest on the idea of a cascade (unlike common multifractal models of classical turbulence).

To avoid insisting on the idea of the Richardson cascade, we removed one sentence at the beginning of the introduction:

In three-dimensional flows, because of the inherently non-linear character of turbulence, energy initially injected at large scales is transferred towards the small scales through a cascade-like process. ~~This cascade idea was first proposed by Richardson and later formalised by Kolmogorov in 1941, leading to his celebrated K41 theory of turbulence.~~

As a result, the final conclusions drawn, which is certainly elegant and nice, is perhaps not as major a breakthrough as the authors envisage it to be. Hence statements about “the existence of a coarse-grained quantity different from ϵ_r ” to capture intermittency while being perhaps true is not as strong a result, in my opinion, to merit publication in Nature Comm. The analogy between classical and quantum turbulence in terms of circulation is certainly clear and compelling but I think the PRX paper already makes this case. The new addition of then using this to understand intermittency in classical turbulence is not, in my opinion, as solid.

The Referee might have misunderstood the final conclusions of our work and failed to appreciate the impact of our results. Our comment “the existence of a coarse-grained quantity different from ϵ_r ”, suggested by our findings, opens and proposes new lines of research to understand circulation intermittency, but we of course have not provided the final answer to the problem of intermittency. We consider that our manuscript contains strong results and new ideas confirmed by numerical data, some of them discussed in our answer to Remark 2, which we believe make this manuscript suitable for Nature Communications.

Remark 3

Specifically,

1. The adaptation of K62/She-Leveque/dissipation-multifractal ideas while being nice stops short of saying where exactly the nature of vortices (in the light of the authors’ work) comes in. I know one can reinterpret the “dimensional” parameter D in such models in the spirit of dimensions of the vortical fields but this is very different from the sharper definition of quantized vortices in quantum turbulence.

Indeed, the parameter D is at the basis of the multi-fractal theory and, as a matter of fact, it is identified with the dimension of the most contributing vortical structures. The notion of fractal dimension depends on the scale of observation. Here, we are looking at scales much larger than the inter-vortex distance at which individual vortex contributions add up. At such scales, the fractal dimensions of the resulting vortical structures are not “sharp” and far from being trivial. To directly measure such a fractal dimensions will require much larger numerical simulations than the one presented in our work, which is very challenging even with modern computational resources.

In our manuscript, we argue that isolated vortex filaments (of dimensionality $D = 1$), while having a strong signature on extreme events in terms of velocity increments, are never responsible for extreme circulation events. In the “ideal” framework of discrete vortex filaments with quantised circulation considered in our work, the velocity is indeed singular at vortex locations, with a velocity field that diverges as $v \sim r^{-1}$ with the distance r to the vortex. This means that an isolated vortex filament *will always contribute* with events of extremely large velocity increments, no matter the considered scale. In our interpretation, this is the very principle of approaches such as the original She–Lévêque model. On the other hand, the same vortex filament will barely contribute to the statistics of circulation (and will definitely not contribute as an extreme event), as the circulation will either be 0 or $\pm\kappa$ depending on the chosen path. This is why, in the interpretation that we propose, the sharp dimensionality $D = 1$ that perfectly describes the signature of intermittency on velocity increment statistics, is irrelevant when one considers circulation statistics.

Remark 4

2. The inhomogeneity of the spatial distribution of vortices in the quantum problem is an important input in this story and its understanding by contrasting the toy model and the GP simulations is an excellent step even if its not for understandable reasons totally rigorous. For classical turbulence, identifying such “filaments” is not easy. Often people resort to QR plots etc as a result. Therefore here again the connections between classical and quantum turbulence is far from straightforward to settle the questions in a definite manner.

We agree that identifying vortex filaments in classical turbulence is not easy. This is one of our motivations for considering quantum fluids as a proxy to classical turbulence. In the spirit of bridging both systems, it would indeed be very interesting to devise new schemes to identify vortex “filaments” in classical flows, in a way that they share the same statistical properties as vortices in quantum fluids.

Note that in our work, we use a well-known model of classical turbulence (She–Lévêque) to predict the spatial distribution of quantum vortices. Doing so, we show how theories, models and knowledge of classical turbulence can be used in quantum turbulence and vice versa.

We hope that the Referee will take the above points regarding the novelty of our present work into account, and reconsider their appreciation of our manuscript.

REVIEWERS' COMMENTS

Reviewer #1 (Remarks to the Author):

The authors have satisfactorily addressed my remarks.

I still believe that the manuscript present results that might be valuable for readers familiar with turbulence modelling, especially those interested in the description of turbulence intermittency. Additionally, I agree that the present results might eventually lead to a better understanding of fluid turbulence in general.

In particular, I appreciate that, in the revised manuscript, the results' relevance for flows that can be experimentally probed is discussed more clearly than in the previous version.

However, in this regard, I want to say here that, in my view, the authors of reference 55 did not succeed in reconstructing Eulerian velocity fields from Lagrangian particle tracking measurements, as claimed in the present manuscript, on lines 607 to 610. In my opinion, they merely treated extremely sparse Lagrangian data as if they were Eulerian data. More importantly, they did not provide any proof that this treatment is correct for the considered data -- the treatment is actually wrong in general, as the authors of the present work should know. In short, I agree that it might be useful to reconstruct Eulerian velocity fields from Lagrangian visualization data but I disagree that this has been already achieved for turbulent flows of superfluid helium-4, at least if one has in mind quantitative studies -- note that the issue was discussed to some extent already by Duda et al. *J. Low Temp. Phys.* 175, 331 (2014) and it is also mentioned by Outrata et al. *J. Fluid Mech.* 924, A44 (2021).

Additionally, in reference 14 the dynamics of individual quantised vortices is not discussed in detail (line 78). In my view, a more quantitative review on the topic is represented by reference 28, or even reference 3.

By the way, which are the 'existent turbulence measurements' mentioned on line 421? Some adequate references should be added at this point.

Reviewer #2 (Remarks to the Author):

I am essentially fine with the authors' reply to my report. They have addressed, through careful discussions, all the points that, in my view, deserved further discussion in their original manuscript, including alternative phenomenological accounts of the scaling properties of turbulent circulation.

The paper adds a meaningful contribution to the literature, not only from its results per se, but also from the fact that it stresses circulation as a fundamental observable to be given more attention in further investigations of turbulent flows.

I have now only a single remark to address (as an optional suggestion to the authors). May be the sentence added to the revised version of the paper,

"It also obscures the possibility of vortex cancellations."

could lead to negative bias to readers, once the model of Apolinário et al. (Ref. [34]) does include, effectively, correlations between positive and negative circulations carried by vortex structures. So, perhaps the above statement could be a bit softened. Referring to the context where the sentence was used, I completely agree with the authors that the interpretation of eddy viscosity as a mean field concept is, at the present moment, just a phenomenological assumption.

Finally, I have greatly appreciated the authors' invitation to discuss the subject of classical and quantum turbulent circulation after the revision process is done.

Reviewer #3 (Remarks to the Author):

The authors essentially improved the paper, taking into account the referee comments, including some of my own ones. I appreciate the authors' efforts to be more accurate in their discussion of the state of the art of theory of turbulence. This lowers their tendency of pulling the blanket over down to an almost acceptable level.

I also appreciate including the energy spectra in Fig. 2, which makes the discussions much more transparent. Now I think that publishing the paper in a specialized journal (possibly with some further improvements) will help the community of the experts in hydrodynamic turbulence to understand this important problem a bit better.

Reviewer #4 (Remarks to the Author):

I have gone through the revised manuscript and the authors' response to all the referee reports. This has helped clarify some of the lingering doubts that I had and I am happy to see this paper published if the Editorially it is judged to meet the standards of the Journal.

Reply to Reviewer #1

We thank once more the Referee for their review and positive comments, and for supporting the publication of our manuscript in Nature Communications. In the following, we provide a point-by-point response for the remarks raised in their second report. Quotes from the Referee’s review are in bold fonts, while relevant modifications to the manuscript are in blue.

Remark 1

The authors have satisfactorily addressed my remarks.

I still believe that the manuscript present results that might be valuable for readers familiar with turbulence modelling, especially those interested in the description of turbulence intermittency. Additionally, I agree that the present results might eventually lead to a better understanding of fluid turbulence in general.

In particular, I appreciate that, in the revised manuscript, the results’ relevance for flows that can be experimentally probed is discussed more clearly than in the previous version.

We thank again the Reviewer for their positive appreciation of our work.

Remark 2

However, in this regard, I want to say here that, in my view, the authors of reference 55 did not succeed in reconstructing Eulerian velocity fields from Lagrangian particle tracking measurements, as claimed in the present manuscript, on lines 607 to 610. In my opinion, they merely treated extremely sparse Lagrangian data as if they were Eulerian data. More importantly, they did not provide any proof that this treatment is correct for the considered data – the treatment is actually wrong in general, as the authors of the present work should know. In short, I agree that it might be useful to reconstruct Eulerian velocity fields from Lagrangian visualization data but I disagree that this has been already achieved for turbulent flows of superfluid helium-4, at least if one has in mind quantitative studies – note that the issue was discussed to some extent already by Duda et al. *J. Low Temp. Phys.* **175**, 331 (2014) and it is also mentioned by Outrata et al. *J. Fluid Mech.* **924**, A44 (2021).

We thank the Reviewer for these precisions on the reconstruction of Eulerian data from Lagrangian particle tracking measurements.

Accordingly, we have modified the referenced paragraph as follows, where we have added the two citations listed by the Reviewer:

[...] Recent experiments have ~~succeeded to reconstruct~~ made initial attempts at reconstructing Eulerian velocity fields from Lagrangian particle tracking measurements in turbulent superfluid helium⁵⁶. Such a technique could be used in principle to measure the velocity circulation in superfluid helium, although addressing high-order statistics might still be challenging. However, note that such an approach is delicate because, due to the two-fluid nature of finite-temperature superfluid helium, particles may fail to capture important Eulerian flow features^{59,60}, and further work is needed to determine its suitability.

Remark 3

Additionally, in reference 14 the dynamics of individual quantised vortices is not discussed in detail (line 78). In my view, a more quantitative review on the topic is represented by reference 28, or even reference 3.

As suggested, we have replaced that reference by Barenghi, L’vov and Roche, PNAS **111**, 4683 (2014) (this is the former reference 28 cited by the Reviewer).

Remark 4

By the way, which are the ‘existent turbulence measurements’ mentioned on line 421? Some adequate references should be added at this point.

After the quoted text, we have added references to relevant experiments and of papers where the resulting experimental data is compared to the She–Lévêque model:

- F. Anselmet, Y. Gagne, E. J. Hopfinger, and R. A. Antonia, High-order velocity structure functions in turbulent shear flows, *J. Fluid Mech.* **140**, 63 (1984).
- R. Benzi, S. Ciliberto, R. Tripiccion, C. Baudet, F. Massaioli, and S. Succi, Extended self-similarity in turbulent flows, *Phys. Rev. E* **48**, R29 (1993).
- Z.-S. She and E. Lévêque, Universal Scaling Laws in Fully Developed Turbulence, *Phys. Rev. Lett.* **72**, 336 (1994).
- G. Boffetta, A. Mazzino, and A. Vulpiani, Twenty-five years of multifractals in fully developed turbulence: A tribute to Giovanni Paladin, *J. Phys. A* **41**, 363001 (2008).

Reply to Reviewer #2

We thank once more the Referee for their review and positive comments, and for supporting the publication of our manuscript in Nature Communications. In the following, we provide a point-by-point response for the remarks raised in their second report. Quotes from the Referee’s review are in bold fonts, while relevant modifications to the manuscript are in green.

Remark 1

I am essentially fine with the authors’ reply to my report. They have addressed, through careful discussions, all the points that, in my view, deserved further discussion in their original manuscript, including alternative phenomenological accounts of the scaling properties of turbulent circulation.

The paper adds a meaningful contribution to the literature, not only from its results per se, but also from the fact that it stresses circulation as a fundamental observable to be given more attention in further investigations of turbulent flows.

We thank again the Reviewer for their positive appreciation of our work.

Remark 2

I have now only a single remark to address (as an optional suggestion to the authors). May be the sentence added to the revised version of the paper,

“It also obscures the possibility of vortex cancellations.”

could lead to negative bias to readers, once the model of Apolinário et al. (Ref. [34]) does include, effectively, correlations between positive and negative circulations carried by vortex structures. So, perhaps the above statement could be a bit softened. Referring to the context where the sentence was used, I completely agree with the authors that the interpretation of eddy viscosity as a mean field concept is, at the present moment, just a phenomenological assumption.

We apologise to the Reviewer, as we did not mean the quoted sentence to lead to negative bias of the cited works.

We have modified the quoted text as follows:

Note that this phenomenological approach mixes a mean-field approximation for determining ν_r with the fluctuations arising from $\varepsilon_r^{1/2}$. ~~It also obscures the possibility of vortex cancellations.~~ **Moreover, in its present form, it does not directly account for vortex cancellations.** Nevertheless, when combined with the standard She–Lévêque model (with $D_\infty = 1$), this model provides an expression for the exponents λ_p as accurate as our mOK62 model in the turbulent case.

Remark 3

Finally, I have greatly appreciated the authors' invitation to discuss the subject of classical and quantum turbulent circulation after the revision process is done.

We thank again the Reviewer, and we will be happy to discuss these subjects after the end of the revision process.

Reply to Reviewer #3

We thank once more the Referee for their review. We acknowledge in particular their previous comments that lead, in our opinion, to important improvements in the new revision of our manuscript.

Remark 1

The authors essentially improved the paper, taking into account the referee comments, including some of my own ones. I appreciate the authors' efforts to be more accurate in their discussion of the state of the art of theory of turbulence. This lowers their tendency of pulling the blanket over down to an almost acceptable level.

We are glad to know that the Reviewer considers that the paper has been improved in its latest revision.

Remark 2

I also appreciate including the energy spectra in Fig. 2, which makes the discussions much more transparent. Now I think that publishing the paper in a specialized journal (possibly with some further improvements) will help the community of the experts in hydrodynamic turbulence to understand this important problem a bit better.

We thank the Reviewer for suggesting the inclusion of the energy spectra in the manuscript.

Reply to Reviewer #4

I have gone through the revised manuscript and the authors' response to all the referee reports. This has helped clarify some of the lingering doubts that I had and I am happy to see this paper published if the Editorially it is judged to meet the standards of the Journal.

We thank once more the Referee for their review and for supporting the publication of our manuscript in Nature Communications.